# LONG-TIME ASYMPTOTICS OF NOISY SVGD OUTSIDE THE POPULATION LIMIT

**V. Priser**
Télécom Paris

**P. Bianchi**
Télécom Paris

**A. Salim**
Microsoft Research

## ABSTRACT

Stein Variational Gradient Descent (SVGD) is a widely used sampling algorithm that has been successfully applied in several areas of Machine Learning. SVGD operates by iteratively moving a set of $n$ interacting particles (which represent the samples) to approximate the target distribution. Despite recent studies on the complexity of SVGD and its variants, their long-time asymptotic behavior (i.e., after numerous iterations $k$) is still not understood in the finite number of particles regime. We study the long-time asymptotic behavior of a noisy variant of SVGD. First, we establish that the limit set of noisy SVGD for large $k$ is well-defined. We then characterize this limit set, showing that it approaches the target distribution as $n$ increases. In particular, noisy SVGD avoids the variance collapse observed for SVGD. Our approach involves demonstrating that the trajectories of noisy SVGD closely resemble those described by a McKean-Vlasov process.

## 1 INTRODUCTION

Sampling is a fundamental task in machine learning, central to Bayesian inference and generative modeling. Mathematically, the task of sampling can be formulated as generating samples, *i.e.*, random variables, from a given (or learned) probability distribution $\pi$. This can be accomplished using a sampling algorithm that iteratively generates samples intended to asymptotically approximate the target distribution.

The question of convergence in distribution of the samples to the target distribution $\pi$ is therefore of primary interest in the theory of sampling. This question has been investigated in several works within the sampling literature, with precise convergence rates established for certain algorithms, such as the celebrated Langevin algorithm. For an overview, see Chewi (2023).

Stein Variational Gradient Descent (SVGD) (Liu & Wang, 2016) is an algorithm for sampling from a target distribution $\pi$ whose density with respect to the Lebesgue measure is known up to a normalizing constant and is written in the form

$$\pi(x) \propto \exp(-F(x)), \quad \text{where} \quad F : \mathbb{R}^d \to \mathbb{R}.$$

SVGD (and its variants) is an alternative to the Langevin algorithm and has been successfully applied in various areas of machine learning; see Liu et al. (2017); Zhang et al. (2018; 2019); Tao et al. (2019); Pu et al. (2017); Kassab & Simeone (2020); Messaoud et al. (2024), among others. For example, the SVGD dynamics can be viewed as a 'kernelized' version of the probability flow ODE used in generative modeling (Song et al., 2020; Chen et al., 2024). The SVGD algorithm takes the form of an interacting particles system with $n$ particles. The empirical distribution of the $n$ particles at iteration $k$, denoted by $\mu_k^n$, is designed to approximate the target distribution $\pi$ as the number of iterations $k$ becomes large.

### 1.1 RELATED WORKS

Several works have investigated the convergence of SVGD, specifically the convergence of $\mu_k^n$ to the target distribution $\pi$.

Most of these works have considered the hypothetical regime $n = \infty$, referred to as the population limit (Korba et al., 2020; Salim et al., 2022; Sun et al., 2023; Nüsken & Renger, 2021). More precisely,

in the population limit, Korba et al. (2020); Salim et al. (2022); Sun et al. (2023) demonstrated that for every $k > 0$,

$$\mathcal{I}_{\text{stein}}(\mu_k^\infty || \pi) < \frac{C}{k}, \tag{1}$$

where $C > 0$ is a constant, and $\mathcal{I}_{\text{stein}}$ denotes the Stein Fisher Information, which measures the discrepancy between the current iterate $\mu_k^\infty$ and the target distribution $\pi$. The convergence in distribution of SVGD to the target $\pi$ in the population limit can be deduced by letting $k \to \infty$ in (1), see Salim et al. (2022).

More recently, several works have considered the finite particles regime $n < \infty$ (Shi & Mackey, 2024; Das & Nagaraj, 2024; Carrillo & Skrzeczkowski, 2023; Liu et al., 2024; Karimi et al., 2023). In this regime, it has been shown that SVGD approximates its population limit provided that $k$ is sufficiently small(Korba et al., 2020; Shi & Mackey, 2024; Lu et al., 2019; Liu, 2017). Combining this result with (1), Shi & Mackey (2024); Carrillo & Skrzeczkowski (2023) demonstrated that $\mathcal{I}_{\text{stein}}(\mu_k^n || \pi) < \frac{C'}{k}$, where $C' > 0$ is a constant, provided that $k$ is small enough (e.g., $k < \log \log(n)$ in Shi & Mackey (2024)). The recent preprint by Balasubramanian et al. (2024), which is concomitant to this paper, presents a similar result to the previously cited works, offering an improved bound. Due to this upper bound on $k$, the convergence of SVGD in the finite particles regime cannot be established by letting $k \to \infty$.

Indeed, SVGD does not converge to the target distribution when $n < \infty$. This is because the iterates of SVGD are discrete measures with a finite support of $n$ points, whereas the target $\pi$ has a continuous density with respect to the Lebesgue measure. Therefore, we pose the following question:

What does SVGD converge to (i.e., as $k \to \infty$) in the finite particles regime (i.e., when $n < \infty$ is fixed)?

To the best of our knowledge, this question remains unanswered, except in the specific case where $\pi$ is a centered Gaussian distribution (see Liu et al. (2024, Theorem 10)). For a fixed $n$, Karimi et al. (2023) demonstrates that SVGD converges in expectation to a system of $n$ continuous-time particles, but this result does not establish consistency with the target distribution $\pi$ as $n$ becomes large.

Nevertheless, we can already make a few observations.

- As mentioned above, SVGD does not converge to the target distribution $\pi$ because the SVGD iterates are discrete, while $\pi$ has a continuous density.

- The best outcome we can generally expect is for the SVGD iterates to converge to some "limit" distribution $\mathcal{L}^n$, which approaches $\pi$ as $n$ increases.

- Even if we were able to demonstrate that the limit $\mathcal{L}^n$ is well-defined (a non-trivial task, as some particles could diverge, for instance), whether $\mathcal{L}^n$ approaches the target distribution $\pi$ as $n$ grows remains an open question. This question is challenging because, empirically, SVGD has been shown not to converge to the target $\pi$ in high-dimensional settings when $n$ is not too large. Specifically, SVGD has been observed to underestimate the variance of the target distribution, and the particles tend to collapse to certain modes of the distribution, see Ba et al. (2021); Zhuo et al. (2018); D'Angelo & Fortuin (2021).

## 1.2 CONTRIBUTIONS

In this paper, we introduce a new noisy variant of SVGD, where each iteration is regularized by noise in the form of an iteration of the Langevin algorithm. We study the "limit" $\mathcal{L}^n$ of our algorithm, noisy SVGD (NSVGD), with $n < \infty$ particles as the number of iterations $k \to \infty$. More precisely, our contributions are as follows:

- We propose a novel noisy variant of SVGD, where each iteration is regularized by noise in the form of an iteration of the Langevin algorithm.

- First, we show that when the number of particles $n < \infty$ is fixed, NSVGD converges to a well-defined limit set $\mathcal{L}^n$ as $k \to \infty$ (Th. 1).

- Next, we describe this limit set $\mathcal{L}^n$: while it does not contain the target $\pi$, we demonstrate that $\mathcal{L}^n$ approaches $\pi$ as $n$ increases (Th. 2).

- Finally, we establish Cor. 1 on the convergence of NSVGD in the regime $\lim_{n\to\infty}\lim_{k\to\infty}$. Since convergence in the regime $\lim_{k\to\infty}\lim_{n\to\infty}$ can be derived from existing works, Cor. 1 implies that the limits $\lim_{n\to\infty}$ and $\lim_{k\to\infty}$ can be interchanged.

- Our approach is based on proving that the trajectories of NSVGD mimic those of a McKean-Vlasov process (Bianchi et al., 2024), a result of independent interest (Proposition 3). It consists of showing that the limit of $\mathscr{L}^n$, as $n$ grows, is a stationary measure of a McKean-Vlasov process. For the pure SVGD algorithm, the target distribution $\pi$ is included in the set of stationary distributions. The noisy variant we propose ensures that $\pi$ is the unique stationary distribution of the associated McKean-Vlasov process. Thus, we are able to show that $\mathscr{L}^n$ converges to $\pi$.

- NSVGD avoids the variance collapse observed in SVGD, a fact we verify experimentally by comparing NSVGD to SVGD (Fig. 1).

## 1.3 PAPER STRUCTURE

This paper is organized as follows. We review some background material in Section 2. In Section 3, we introduce our main algorithm, NSVGD. Next, we present our main results regarding the convergence of NSVGD in Section 4. In Section 5, we provide a sketch of our convergence proof, which relies on relating the trajectories of NSVGD to those of a McKean-Vlasov process. In Section 6, we empirically demonstrate that NSVGD, unlike SVGD, does not suffer from particles collapse. Finally, we conclude in Section 7. The proofs are deferred to the Appendix.

## 2 BACKGROUND

### 2.1 NOTATIONS

The Euclidean inner product and norm of $\mathbb{R}^d$ are denoted $\langle\cdot,\cdot\rangle$ and $\|\cdot\|$. We consider a Reproducing Kernel Hilbert Space (RKHS) $\mathcal{H}_0$ whose kernel is denoted $K : \mathbb{R}^d \times \mathbb{R}^d \to \mathbb{R}$. For $x, y \in \mathbb{R}^d$, we denote by $\nabla_y K(x,y)$ the gradient of $K$ with respect to $y$. The product space $\mathcal{H} := \mathcal{H}_0^d$, is a Hilbert space whose inner product and norm are denoted $\langle\cdot,\cdot\rangle_{\mathcal{H}}$ and $\|\cdot\|_{\mathcal{H}}$. We denote by $[n]$ the set of integers $\{1,\ldots,n\}$. We say that a quantity $\ell_t^n$ converges as $(t,n) \to (\infty,\infty)$ in some sense to $\ell$ if, for every sequence $(t_n,\varphi_n) \to (\infty,\infty)$, $\ell_{t_n}^{\varphi_n}$ converges as $n \to \infty$ to $\ell$.

### 2.2 OPTIMAL TRANSPORT

For every topological space $E$, we denote by $\mathcal{P}(E)$ the set of probability measures on the Borel $\sigma$-field $\mathcal{B}(E)$. If $E$ is a Polish (complete, metrizable) space, then $\mathcal{P}(E)$ equipped with the weak$\star$ topology is Polish as well. A subset $\mathcal{A}$ of random variables on $E$ is called *tight*, if, for every $\varepsilon > 0$, there exists a compact set $A \subset E$, such that $\mathbb{P}(X \in A) > 1 - \varepsilon$, for every $X \in \mathcal{A}$. If $E$ is a Banach space, we define

$$\mathcal{P}_2(E) := \left\{\mu \in \mathcal{P}(E) \ : \ \int \|x\|^2 d\mu(x) < \infty\right\},$$

and the Wasserstein-2 distance by

$$W_2(\mu,\nu) := \left(\inf_{\varsigma\in\Pi(\mu,\nu)} \int \|x-y\|^2 d\varsigma(x,y)\right)^{1/2},$$

where $\Pi(\mu,\nu)$ is the set couplings of $\mu \in \mathcal{P}_2(E)$ and $\nu \in \mathcal{P}_2(E)$, *i.e.*, the set of measures $\varsigma \in \mathcal{P}(E \times E)$ such that $\varsigma(\cdot \times E) = \mu$ and $\varsigma(E \times \cdot) = \nu$. The Wasserstein space, *i.e.*, the set $\mathcal{P}_2(E)$ endowed with the distance $W_2$, is a Polish space.

In the proofs, we need to consider the case where the space $E$ coincides with the set $\mathcal{C}$ of continuous function on $[0,\infty)$ to $\mathbb{R}^d$. Eventhough $\mathcal{C}$ is not a Banach space, the definitions follow the same lines. The set $\mathcal{C}$ is equipped with the topology of uniform convergence on compact intervals. For every $\rho \in \mathcal{P}(\mathcal{C})$, we denote by $\rho^T$ the restriction of $\rho$ to functions on the compact interval $[0,T]$ (that is, $\rho^T = (\pi_{[0,T]})_\# \rho$, the pushforward of $\rho$ by the map $\pi_{[0,T]}$ which, to every function $f \in \mathcal{C}$, associates its restriction to the compact interval $[0,T]$). We denote by $\mathcal{P}_2(\mathcal{C})$ the set of measures

$\rho \in \mathcal{P}(\mathcal{C})$ such that $\rho^T \in \mathcal{P}_2(C([0,T], \mathbb{R}^d))$ for all $T > 0$. This space is naturally equipped with the following topology: a sequence $\rho_n$ converges to $\rho$ in the Wasserstein-2 sense if $\rho_n^T \to \rho^T$ in the Wasserstein-2 sense, for every $T > 0$. Then, $\mathcal{P}_2(\mathcal{C})$ is metrizable, and we denote by $W_2(\rho, \rho')$ a proper distance (Bianchi et al., 2024, Sec. 2.2).

## 2.3 FUNCTIONAL INEQUALITIES

Let $\pi \in \mathcal{P}_2(\mathbb{R}^d)$ be the target distribution, *i.e.*, $\pi \propto \exp(-F)$. The Kullback-Leibler divergence with respect to $\pi$ is defined for every $\mu \in \mathcal{P}_2(\mathbb{R}^d)$ by:

$$D_{\mathrm{KL}}(\mu || \pi) = \int \log \frac{d\mu}{d\pi} d\mu,$$

if $\mu$ has a density $\frac{d\mu}{d\pi}$ w.r.t. $\pi$, and $D_{\mathrm{KL}}(\mu || \pi) = +\infty$ else. The Fisher Information w.r.t. $\pi$ is defined by:

$$\mathcal{I}(\mu || \pi) := \int \left\| \nabla \log \frac{d\mu}{d\pi} \right\|^2 d\mu(x).$$

We recall the Log Sobolev Inequality (LSI) that relates the Kullback-Leibler divergence and the Fisher Information.

*Definition* 1 (Logarithmic Sobolev Inequality). The distribution $\pi$ satisfies the Logarithmic Sobolev Inequality, if there exists $\alpha > 0$ such that for every $\mu \in \mathcal{P}_2(\mathbb{R}^d)$,

$$D_{\mathrm{KL}}(\mu || \pi) \leq \frac{1}{2\alpha} \mathcal{I}(\mu || \pi).$$

The LSI is satisfied when $F$ is $\alpha$-strongly convex but can also be used to study the convergence of sampling algorithms in the case where $F$ is not convex (Villani, 2009, Section 21) (see also Vempala & Wibisono (2019)). Finally, we define the Stein Fisher Information w.r.t. $\pi$ by:

$$\mathcal{I}_{\mathrm{stein}}(\mu || \pi) := \left\| P_\mu \nabla \log \frac{d\mu}{d\pi} \right\|_{\mathcal{H}}^2,$$

where $P_\mu : L^2(\mu) \to \mathcal{H}$ is the so-called kernel integral operator $P_\mu f = \int K(\cdot, y) f(y) d\mu(y)$.

# 3 NOISY STEIN VARIATIONAL GRADIENT DESCENT (NSVGD)

The Stein Variational Gradient Descent (SVGD) algorithm (Liu & Wang, 2016) is used to sample from a distribution $\pi \propto \exp(-F)$, where $F : \mathbb{R}^d \to \mathbb{R}$ is a differentiable function. At every iteration $k$, the algorithm updates the values of $n$ $\mathbb{R}^d$-valued vectors, refered to as the particles $X_k^{1,n}, \cdots, X_k^{n,n}$. We study a generalization of SVGD, called NSVGD, that incorporates noise in the form of a Langevin iteration at each step of SVGD.

Let $(\Omega, \mathcal{F}, \mathbb{P})$ be a probability space, $\lambda \geq 0$ and $(\gamma_k)$ be a positive deterministic sequence in $\mathbb{R}$. Starting with a $n$–uple $(X_0^{1,n}, \ldots, X_0^{n,n})$ of $\mathbb{R}^d$-valued random variables, the particles are updated according to Algorithm 1 where $(\xi_k^{i,n})_{i,k}$ is a family of i.i.d standard Gaussian vectors in $\mathbb{R}^d$.

---

**Algorithm 1** Noisy Stein Variational Gradient Descent (NSVGD)

---

**Initialization**: generate $n$ particles $(X_0^{1,n}, \ldots, X_0^{n,n})$
**for** $k = 0, 1, 2, \ldots$ **do**
    **for** $i = 1, 2, \ldots, n$ **do**

$$X_{k+1}^{i,n} = X_k^{i,n} - \frac{\gamma_{k+1}}{n} \sum_{j \in [n]} \left( K(X_k^{i,n}, X_k^{j,n}) \nabla F(X_k^{j,n}) - \nabla_y K(X_k^{i,n}, X_k^{j,n}) \right)$$
$$\underbrace{-\lambda \gamma_{k+1} \nabla F(X_k^{i,n}) + \sqrt{2\lambda \gamma_{k+1}} \xi_{k+1}^{i,n}}_{\text{Langevin regularization}}. \quad (2)$$

    **end for**
**end for**

---

NSVGD boils down to the standard deterministic SVGD algorithm when $\lambda = 0$. The regularization parameter $\lambda > 0$ introduces noise into the algorithm, which ensures that the set of limiting distributions is unique and coincides with the target. This is a property that the pure SVGD algorithm does not exhibit, as described in the introduction (see Sec. 6 for a more detailed discussion).

**Assumption 1.** *Let the following holds.*

i) $(\gamma_k)$ *is a non-negative deterministic sequence satisfying* $\lim_{k\to\infty} \gamma_k = 0$, *and* $\sum_k \gamma_k = +\infty$.

ii) $(\xi_k^{i,n})_{k\in\mathbb{N}, i\in[n]}$ *is an i.i.d. sequence of standard Gaussian variables, independent of* $(X_0^{i,n})_{i\in[n]}$.

NSVGD allows for the approximation of linear functionals of the form $\int f \, d\pi$, where $f$ is an arbitrary integrand, by the discrete sum $\frac{1}{n}\sum_{i=1}^{n} f(X_k^{i,n})$. The latter can be written as $\int f \, d\mu_k^n$, where $\mu_k^n$ is the empirical measure of the particles, defined by:

$$\mu_k^n := \frac{1}{n} \sum_{i\in[n]} \delta_{X_k^{i,n}} .$$

Note that $(\mu_k^n)_k$ is a sequence of *random* measures. A useful convergence result for NSVGD involves studying the convergence in probability of this sequence towards the target distribution $\pi$. In some situations, it is more convenient to study the *averaged* empirical measure $\bar{\mu}_k^n$, defined for $k, n \in \mathbb{N}^*$ by:

$$\bar{\mu}_k^n := \frac{\sum_{i\in[k]} \gamma_i \mu_i^n}{\sum_{i\in[k]} \gamma_i} .$$

# 4 CONVERGENCE RESULTS OF NSVGD

## 4.1 LIMIT SET OF NSVGD IS WELL-DEFINED

We start our analysis by studying the limit set of NSVGD as $k$ tend to infinity, for a fixed number $n$ of particles. As the number of particles is fixed, it cannot be expected that the limit of $\mu_k^n$ coincides with $\pi$ as $k \to \infty$, because a discrete measure with a fixed number of atoms cannot approach a density.

We begin by stating the assumptions that ensure the stability of our algorithm.

**Assumption 2.** *There exists two non-negative constant* $c, C$, *such that for every* $x, y \in \mathbb{R}^d$, *the following holds.*

i) *The hessian* $\mathbf{H}(F)(x)$ *is well-defined and* $\|\mathbf{H}(F)(x)\|_{op} \leq C$.

ii) $c\|x\|^2 - C \leq \min(\|\nabla F(x)\|^2, |F(x)|)$.

iii) $\|K(\cdot, y)\|_{\mathcal{H}_0} + \|\nabla_y K(\cdot, y)\|_{\mathcal{H}} \leq C$.

iv) $\sup_n \mathbb{E}\left((X_0^{1,n})^4\right) < \infty$.

This assumption is satisfied, for instance, when $\pi$ is a mixture of Gaussians and $K$ is either a Gaussian or a polynomial kernel. Given the previous assumption, we can establish the stability of our algorithm, in the form of the following propposition.

**Proposition 1.** *Let Assumptions 1 and 2 be satisfied. Assume* $\lambda > 0$. *Then,* $\sup_{k,n} \mathbb{E}\|X_k^{1,n}\|^4 < \infty$.

We formally describe the limit set of the empirical measures in a distributional sense.
*Definition* 2 (Distributional limit set). Let $\nu, (\nu_k : k \in \mathbb{N})$ be random variables on $\mathcal{P}(\mathbb{R}^d)$. We say that $\nu$ is a distributional cluster point of $(\nu_k)$, if $\nu_k$ converges in distribution to $\nu$ along a subsequence. The distributional limit set $\mathscr{L}((\nu_k))$ of the sequence $(\nu_k)$ is defined as the set of distributional cluster points of $(\nu_k)$.

We denote by $\mathscr{L}^n := \mathscr{L}((\mu_k^n))$ the distributional limit set of the sequence $(\mu_k^n : k \in \mathbb{N})$, when $k \to \infty$, $n$ being fixed. In words, $\mathscr{L}^n$ is the set of random measures $\nu^n$ such that $\mu_k^n$ converges to $\nu_n$ in distribution, along a subsequence. Similarly, we denote by $\overline{\mathscr{L}}^n$ the limit set of the sequence $(\bar{\mu}_k^n)$.

Prop. 1 is the key component for establishing our first theorem.

**Theorem 1.** *Let Assumptions 1 and 2 hold. Assume $\lambda > 0$. Then, for every $n \in \mathbb{N}^*$, the sequence of random variables $(\mu_k^n)_k$ is tight. As a consequence, the sets $\mathscr{L}^n$ and $\overline{\mathscr{L}}^n$ are non empty. Finally, all random measures of $\mathscr{L}^n$ and $\overline{\mathscr{L}}^n$ belong almost surely to $\mathcal{P}_2(\mathbb{R}^d)$.*

It remains to characterize the limit sets. As mentioned earlier, the random variable equal to $\pi$ a.s. does not belong to the set $\mathscr{L}^n$. Therefore, the question is whether $\mathscr{L}^n$ reduces to the singleton $\pi$ as $n$ goes to infinity.

### 4.2 DESCRIPTION OF THE LIMIT SET

The following assumption is technical and ensures that the limiting measures of $\overline{\mathscr{L}}^n$ and $\mathscr{L}^n$, in the sense of the definition below, admit a density with respect to the Lebesgue measure. In other words, it ensures that the marginals of the McKean-Vlasov distributions in Definition 4 have a density.

**Assumption 3.** *There exists $\beta > 0$, such that for every $x, x', y \in \mathbb{R}^d$, we obtain*

$$|K(x,y) - K(x',y)| + \|\nabla_y K(x,y) - \nabla_y K(x',y)\| \le C \|x - x'\|^\beta .$$

*Definition* 3. For every $n \ge 1$, let $\mathscr{E}^n$ be a set of random measures on $\mathcal{P}_2(\mathbb{R}^d)$. We say that the sequence of random sets $(\mathscr{E}^n : n \in \mathbb{N}^*)$ converges in probability to $\pi$, denoted by $\mathscr{E}^n \xrightarrow{\mathbb{P}} \pi$, if the Hausdorff-Wasserstein distance between $\mathscr{E}^n$ and $\pi$ converges in probability to zero:

$$\forall \varepsilon > 0, \quad \lim_{n \to \infty} \mathbb{P}\big( \sup_{\nu \in \mathscr{E}^n} W_2(\nu, \pi) > \varepsilon \big) = 0 .$$

The following theorem establishes the convergence of the averaged empirical measure, in the sense of Def. 3, to the target by first taking the limit as $k \to \infty$ and then the limit as $n \to \infty$.

**Theorem 2.** *Let Assumptions 1 , 2, and 3 hold. Assume $\lambda > 0$. Then,*

$$\overline{\mathscr{L}}^n \xrightarrow[n \to \infty]{\mathbb{P}} \pi .$$

The motivation for studying the limit set $\overline{\mathscr{L}}^n$ of the averaged measure $\bar{\mu}_k^n$ is technical. However, the same result for the empirical measure $\mu_k^n$ can also be obtained, provided an additional assumption on the target density is satisfied.

**Assumption 4.** *The distribution $\pi$ satisfies the Logarithmic Sobolev Inequality for a constant $\alpha > 0$.*

**Theorem 3.** *Let Assumptions 1 , 2, 3 and 4 hold. Assume $\lambda > 0$. Then,*

$$\mathscr{L}^n \xrightarrow[n \to \infty]{\mathbb{P}} \pi .$$

### 4.3 LONG-TIME CONVERGENCE OF THE EMPIRICAL MEASURE

As a consequence of Th. 2 and Th. 3 respectively, we can characterize the long-time convergence of the empirical measure of the particles, averaged and non-averaged respectively.

**Corollary 1.** *Let Assumptions 1 , 2 and 3 hold. Assume $\lambda > 0$. Then, for every $\varepsilon > 0$,*

$$\lim_{n \to \infty} \limsup_{k \to \infty} \mathbb{P}(W_2(\bar{\mu}_k^n, \pi) > \varepsilon) = 0 .$$

*If Assumption 4 also holds, the same result applies when $\bar{\mu}_k^n$ is replaced by $\mu_k^n$.*

Since the convergence in the regime $\lim_{k \to \infty} \limsup_{n \to \infty}$ can be deduced from the existing works mentioned above, Cor. 1 implies that $\lim_{n \to \infty}$ and $\lim_{k \to \infty}$ can be exchanged.

## 5 SKETCH OF THE CONVERGENCE PROOF OF NSVGD

To briefly explain our proof technique, we first show the stability of our algorithm, which, by a compactness argument, establishes the existence of a limiting distribution (Prop. 2). Secondly, the

limiting distributions have a specific structure: they are solutions to the McKean-Vlasov equation (Prop. 3). Since the limiting distributions are also stationary, we must identify the stationary distribution of the McKean-Vlasov equation. We first show that the Kullback-Leibler (KL) divergence with respect to the target decreases along the trajectories of the McKean-Vlasov solutions (Prop. 4). This implies that the target is the unique stationary distribution of the McKean-Vlasov solution and therefore coincides with the marginal of the limiting distributions.

## 5.1 INTERPOLATED PROCESS

We consider for each $i \in [n]$ the random continuous-time process $\bar{X}^{i,n} : [0, \infty) \to \mathbb{R}^d, t \mapsto \bar{X}^{i,n}_t$ defined as the piecewise linear interpolation of the particles $(X^{i,n}_k)_k$. Specifically, writing $\tau_k := \sum_{j=1}^{k} \gamma_j$ , for each $k \in \mathbb{N}$, we define:

$$\forall t \in [\tau_k, \tau_{k+1}), \quad \bar{X}^{i,n}_t := X^{i,n}_k + \frac{t - \tau_k}{\gamma_{k+1}} \left( X^{i,n}_{k+1} - X^{i,n}_k \right).$$

The interpolated processes $\bar{X}^{i,n}$, for $i \in [n]$, are elements of the set $\mathcal{C}$ of continuous functions on $[0, \infty) \to \mathbb{R}^d$. Rather than solely examining the empirical measure of the particles $X^{i,n}_k$, our approach focuses on analyzing the empirical measure of the interpolated processes $\bar{X}^{i,n}$ across the entire positive real line. Define:

$$m^n_t := \frac{1}{n} \sum_{i=1}^{n} \delta_{\bar{X}^{i,n}_{t+\cdot}} ,$$

for each $n$ and $t$. For a function $x \in \mathcal{C}$, we used the notation $x_{t+\cdot}$ as an element of $\mathcal{C}$ defined by $s \mapsto x_{t+s}$. Note that $m^n_t$ is a random variable on $\mathcal{P}_2(\mathcal{C})$. The empirical measure $\mu^n_k$ of the discrete particles can be deduced from $m^n_t$ by marginalization, which is why we focus on $m^n_t$ from now on.

## 5.2 MCKEAN-VLASOV DISTRIBUTIONS

For a fixed $n$, the particles $X^{i,n}_k$, for $i \in [n]$, can be interpreted as an Euler discretization scheme of a stochastic differential equation involving $n$ continuous-time particles. As the discretization step $\gamma_k$ tends to zero, the interpolated processes eventually share the same behavior as the continuous-time particles as $k$ tends to infinity. Moreover, in the population limit where $n$ is large, any of the continuous-time particles coincides, in law, with the solution to a McKean-Vlasov equation, as defined below. This phenomenon is known as the propagation of chaos. We refer to Chaintron & Diez (2022) for a detailed exposition.

*Definition* 4. We say that a measure $\rho \in \mathcal{P}_2(\mathcal{C})$ is a McKean-Vlasov distribution, if it coincides with the pathwise law of a weak solution $(X_t)_{t \geq 0}$ to the nonlinear Stochastic Differential Equation (SDE)

$$dX_t = - \int \left( K(X_t, y) \nabla F(y) - \nabla_y K(X_t, y) \right) d\rho_t(y) \, dt - \lambda \nabla F(X_t) \, dt + \sqrt{2\lambda} \, dW_t,$$

where $(W_t)_{t \geq 0}$ is a standard Brownian motion. Denote by $\mathbb{V}_2$ the set of McKean-Vlasov distributions.

## 5.3 LIMIT MEASURES OF NSVGD ARE MCKEAN-VLASOV DISTRIBUTIONS

It remains to explain in which sense, the empirical measures $m^n_t$ converge to a McKean-Vlasov distribution as $(t, n) \to (\infty, \infty)$. The question requires the introduction of the following measure:

$$M^n_t := \frac{1}{t} \int_0^t \delta_{m^n_s} ds \,.$$

To summarize, we introduced the following of random variables: (process level) $\bar{X}^{i,n}$ is a r.v. on $\mathcal{C}$; (process-measure level) $m^n_t$ is a r.v. on $\mathcal{P}_2(\mathcal{C})$; (process-measure-measure level) $M^n_t$ is a r.v. on $\mathcal{P}(\mathcal{P}_2(\mathcal{C}))$. As a consequence of Prop. 1, we obtain the following result.

**Proposition 2.** *Let Assumptions 1 and 2 be satisfied. Assume $\lambda > 0$. For every $n \in \mathbb{N}^*$, the sequence of random variables $(M^n_t)_t$ is tight.*

In particular, Proposition 2 implies Th. 1 and the fact that the limit set of SVGD is non-empty. It remains to characterize the latter in the doubly asymptotic regime where $t, n$ both tend to infinity. To that end, we study the (distributional) limit points of $(M_t^n)$, as $(t, n) \to (\infty, \infty)$. The following result is a extracted from Bianchi et al. (2024, Lem. 9).

**Proposition 3.** *Let Assumptions 1 and 2 be satisfied. Assume $\lambda > 0$. Let $M$ be a random measure on $\mathcal{P}(\mathcal{P}_2(\mathcal{C}))$ such that $M_t^n$ converges in distribution to $M$ as $(t, n) \to (\infty, \infty)$, along some subsequence. Then, $M(\mathbb{V}_2) = 1$ a.s.*

Let us explain the main consequence of this result. Let $f$ be the function defined by $f(\rho) = W_2(\rho, \mathbb{V}_2)$ for every $\rho \in \mathcal{P}_2(\mathbb{R}^d)$. When $M_t^n$ tends to $M$ in distribution along some subsequence, our definition of $M_t^n$ implies that:

$$\int f dM_t^n = \frac{1}{t} \int_0^t W_2(m_s^n, \mathbb{V}_2) ds \xrightarrow{\mathcal{D}} \int W_2(\rho, \mathbb{V}_2) dM(\rho) = 0 \,,$$

where the symbol $\xrightarrow{\mathcal{D}}$ stand for convergence in distribution. This shows that, in an ergodic sense, $m_t^n$ converges in probability to the set of McKean-Vlasov distributions, as $(t, n) \to (\infty, \infty)$.

### 5.4 LIMIT MEASURES OF NSVGD ARE TIME-SHIFT RECURRENT

More can be said about the particular McKean-Vlasov distribution in the limit set. For every $\tau > 0$, denote by $\Phi_\tau : \mathcal{P}(\mathcal{C}) \to \mathcal{P}(\mathcal{C})$ the map which shifts a process-measure by a time $\tau$, namely, $\Phi_\tau(\rho) : f \mapsto \int f(x_{\tau+\cdot}) d\rho(x)$. Obviously, $\Phi_\tau(m_t^n) = m_{\tau+t}^n$, which in turn implies that, as $t \to \infty$, for every bounded continuous function $G : \mathcal{P}(\mathcal{C}) \to \mathbb{R}$,

$$\int G(\Phi_\tau(\rho)) dM_t^n(\rho) = \frac{1}{t} \int_0^t G(m_{\tau+s}^n) ds \simeq \frac{1}{t} \int_0^t G(m_s^n) ds = \int G(\rho) dM_t^n(\rho) \,,$$

where the precise statement is found in the supplementary (see also Bianchi et al. (2024, Lem. 10)). Passing to the limit, this implies that every distributional limit point $M$ of $M_t^n$ is shift-invariant, in the sense that $\int G \circ \Phi_\tau dM = \int G dM$ a.s., for every bounded continuous $G$ and every $\tau > 0$.

### 5.5 CONVERGENCE OF NSVGD TO THE TARGET

For any process-measure $\rho \in \mathcal{P}(\mathcal{C})$, we denote by $(\rho_t : t \geq 0)$ its marginals in $\mathcal{P}(\mathbb{R}^d)$.

**Proposition 4.** *Let Assumption 2 and 3 hold. Assume $\lambda > 0$. Let $t_2 > t_1 > 0$. For every $\rho \in \mathbb{V}_2$ and every $t \in [t_1, t_2]$, $\rho_t$ admits a differentiable density w.r.t. the Lebesgue measure. Moreover,*

$$D_{\mathrm{KL}}(\rho_{t_2}||\pi) - D_{\mathrm{KL}}(\rho_{t_1}||\pi) = - \int_{t_1}^{t_2} \left( \mathcal{I}_{stein}(\rho_t||\pi) + \lambda \mathcal{I}(\rho_t||\pi) \right) dt \,.$$

We are now able to obtain Th. 2. Let $M \in \mathcal{P}(\mathcal{P}(\mathcal{C}))$ be the random variable given by Prop. 3. Recall that by Sec. 5.4, the measure $M$ is time-shift recurrent. Hence, putting aside technical details, for every $t > 0, \tau > 0$, almost surely, we obtain

$$\int D_{\mathrm{KL}}(\rho_t||\pi) dM(\rho) = \int D_{\mathrm{KL}}(\rho_{t+\tau}||\pi) dM(\rho) \,.$$

By Prop. 3 and 4, this implies $\iint_t^{t+\tau} \left( \mathcal{I}_{stein}(\rho_s||\pi) + \lambda \mathcal{I}(\rho_s||\pi) \right) ds dM(\rho) = 0$ . The l.h.s. of this equation is zero if and only if $\rho = \pi$ for every $\rho$ in the support of $M$. Thus, in an ergodic sense, the marginals of the measure $m_t^n$ converges in probability to $\pi$, as $(t, n) \to (\infty, \infty)$ (see Prop. 7 in the Appendix). Leveraging some technical details, this in turn yields Th. 2.

The last step is to establish Th. 3 under the additional Assumption 4. In other words, one should discard the time-averaging. This can be done in the situation where, as $t \to \infty$, the marginal $\rho_t$ of any McKean-Vlasov distribution $\rho \in \mathbb{V}_2$ converges to $\pi$ uniformly in the initial point $\rho_0$ in a compact set. This can be established using the LSI, as shown by the following result.

**Proposition 5.** *Let the assumptions of Prop. 4 hold. Moreover, we assume that Assumption 4 is satisfied with $\alpha > 0$ and $\lambda > 0$. For any compact set $\mathcal{K} \subset \mathcal{P}_2(\mathcal{C})$, for every $t_2 > t_1 > 0$, there exists a constant $C_{t_1, \mathcal{K}} > 0$ depending on $t_1$ and $\mathcal{K}$, such that*

$$\sup_{\rho \in \mathbb{V}_2 \cap \mathcal{K}} W_2(\rho_{t_2}, \pi) \leq C_{t_1, \mathcal{K}} \mathrm{e}^{-\alpha \lambda (t_2 - t_1)} \,.$$

## 6 NSVGD AVOIDS THE PARTICLES COLLAPSE

The convergence results of Sec. 4 show the convergence of NSVGD in a doubly asymptotic regime $(k, n) \to (\infty, \infty)$. These convergence results could be reproduced for the deterministic SVGD algorithm. However, in the case of SVGD, our approach would show the convergence of SVGD to a set that includes the target $\pi$, but can also include Dirac measures at stationary points of $F$. Indeed, the McKean-Vlasov process of SVGD (*i.e.*, the case $\lambda = 0$) is stationary at $\delta_x$ for any $x \in \mathbb{R}^d$ such that $\nabla F(x) = 0$ and $\nabla_y K(x, x) = 0$[1].

This observation is in line with empirical results showing that the deterministic SVGD algorithm may not converge in high dimensions and instead collapse to some Diracs, which represent modes of the target distribution (Ba et al., 2021; Zhuo et al., 2018; D'Angelo & Fortuin, 2021). Specifically, Ba et al. (2021) shows that variance collapse occurs for SVGD in the regime when $d/n > 1$. We showed (Th.2 and3) that NSVGD converges to the target and, in particular, does not collapse to Dirac measures. Our theoretical results are given for a fixed dimension $d$ and cannot be computed uniformly in the dimension. Therefore, we show experimentally that NSVGD, in the setup of Ba et al. (2021) ($d/n > 1$), does not exhibit variance collapse.

Fig. 1 (see Appendix for larger figures) reproduces an experiment from Ba et al. (2021) on the variance collapse of SVGD. We added our algorithm, NSVGD, to the plot. In Fig. 2, we show that the collapse occurs even when the number of particles $n$ is large compared to the dimension $d$.

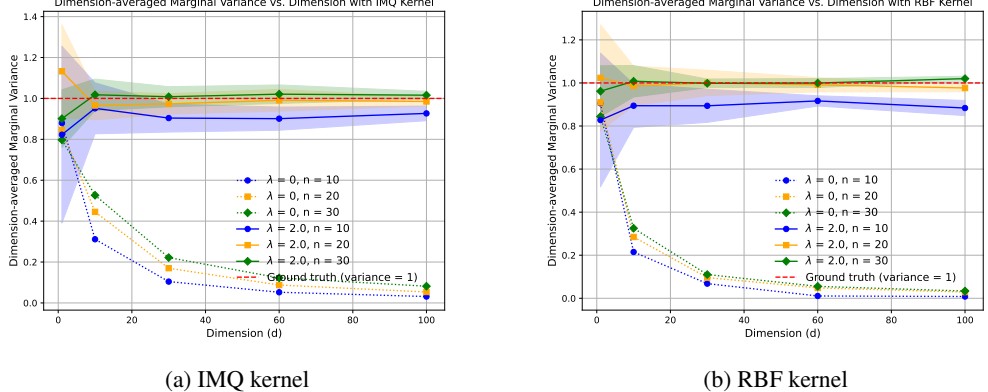

(a) IMQ kernel          (b) RBF kernel

Figure 1: Dimension-averaged Marginal Variance of SVGD and NSVGD at convergence for sampling from a standard Gaussian.

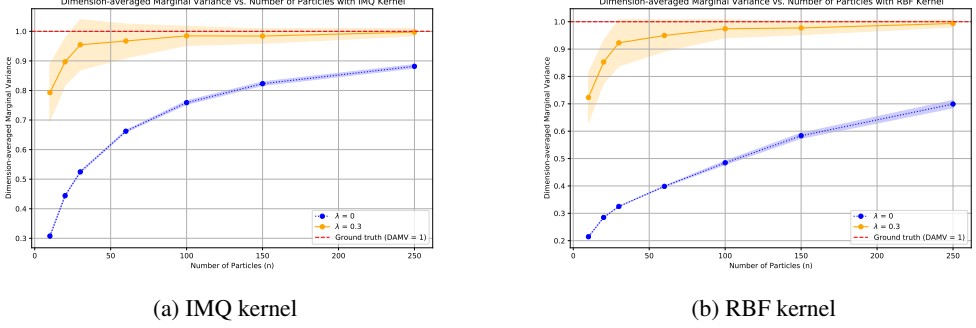

(a) IMQ kernel          (b) RBF kernel

Figure 2: Dimension-averaged Marginal Variance of SVGD and NSVGD at convergence for sampling from a standard Gaussian of fixed dimension $d = 10$.

---

[1]On the contrary, every stationary distribution of the McKean-Vlasov process of NSVGD (*i.e.*, the case $\lambda > 0$) must have a density w.r.t. Lebesgue thanks to the noise injection.

**The setup is the following.** We consider the task of sampling from a standard Gaussian with NSVGD and SVGD. We use the two most standard kernels for running SVGD: the Radial Basis Function (RBF) kernel, a.k.a. Gaussian kernel $K(x,y) = \exp(-\frac{1}{2}\|x-y\|^2)$ and the Inverse Multi-Quadratic (IMQ) kernel (Gorham & Mackey, 2017; Kanagawa et al., 2022) $K(x,y) = \frac{1}{\sqrt{1+\frac{1}{2}\|x-y\|^2}}$. We simulate NSVGD until convergence (*i.e.*, after a large number $k = 200$ of iterations) for different values of the dimension $d$, the number of particles $n$, and the regularization parameter $\lambda$. When $\lambda = 0$, NSVGD boils down to the deterministic SVGD. The particles are initialized randomly from a standard Gaussian and the step size is set to $\gamma_k = 10/k$.

Given a probability distribution over $\mathbb{R}^d$, the Dimension-Averaged Marginal Variance (DAMV) is a statistics of the distribution equal to the average across the $d$ coordinates of the variance of each coordinate. We reproduce an experiment from Ba et al. (2021) where they plotted the DAMV of SVGD after a large number of iterations against the dimension. We added NSVGD to the plot, see Fig. 1 and 2. Since NSVGD is random, its DAMV is a random number, therefore we plotted the averaged value of the DAMV over 10 runs and represented the standard deviation of the DAMV in the shaded area behind the curve. Our Python script is available in the Supplementary Material and Fig. 1 and 2 are available in the Appendix in a larger format.

From Fig. 1 and 2, two important observations can be made:

- Since each point in the figure represents a statistical measure (the DAMV) for NSVGD after numerous iterations, our theoretical analysis predicts that as $n$ increases, the DAMV values for NSVGD should converge to the DAMV of the standard Gaussian, which is 1. This convergence towards 1 with increasing $n$ is indeed what we observe in the NSVGD data.

- Contrasting this, SVGD shows a different behavior where its DAMV tends to zero as the dimension increases, as discussed in Ba et al. (2021). Unlike SVGD, NSVGD does not exhibit this variance collapsing behavior.

**The Langevin regularization** We studied a mixture of the Langevin algorithm and the SVGD algorithm. We have shown that the Langevin component is useful both in practice and in theory. However, the Langevin part does not need to be large. SVGD has demonstrated superiority in several applications (faster convergence, adaptability, etc.). By setting $\lambda$ sufficiently small, we retain the advantages of the SVGD algorithm while ensuring convergence guarantees. The optimal balance between Langevin and SVGD (in terms of the optimal convergence rate of the Wasserstein distance between the empirical measure and the target distribution) is application-dependent, and a detailed study of this balance is left for future work. In Sec. C, we compare NSVGD with the Langevin and SVGD algorithms for various values of $\lambda$.

## 7 CONCLUSION

What does a user do? A user sets a finite value for the number $n$ of particles and then runs the algorithm until convergence. Therefore, understanding what the algorithm converges to when $n$ is finite is of primary importance. In this work, we provided insights into the limit set $\mathscr{L}^n$ of NSVGD after a large number of iterations. We showed that this limit set is well-defined and approaches the target as $n$ increases. Several conclusions follow from these results. In particular, NSVGD, unlike SVGD, provably avoids collapsing to certain modes of the target distribution.

Our work raises several questions regarding the convergence speed of NSVGD. First, can we quantify the convergence of NSVGD to the set $\mathscr{L}^n$? Then, can we quantify the convergence of the set $\mathscr{L}^n$ to the target? Finally, how should the regularization parameter $\lambda$ be chosen, and what is its effect on the convergence rate?

These problems, which are not addressed in the existing literature on SVGD and its variants, would deepen our understanding of interacting particles systems for sampling, in a regime that matters from a practical perspective.

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

# Appendix

## CONTENTS

# A  Fig. 1 in larger format

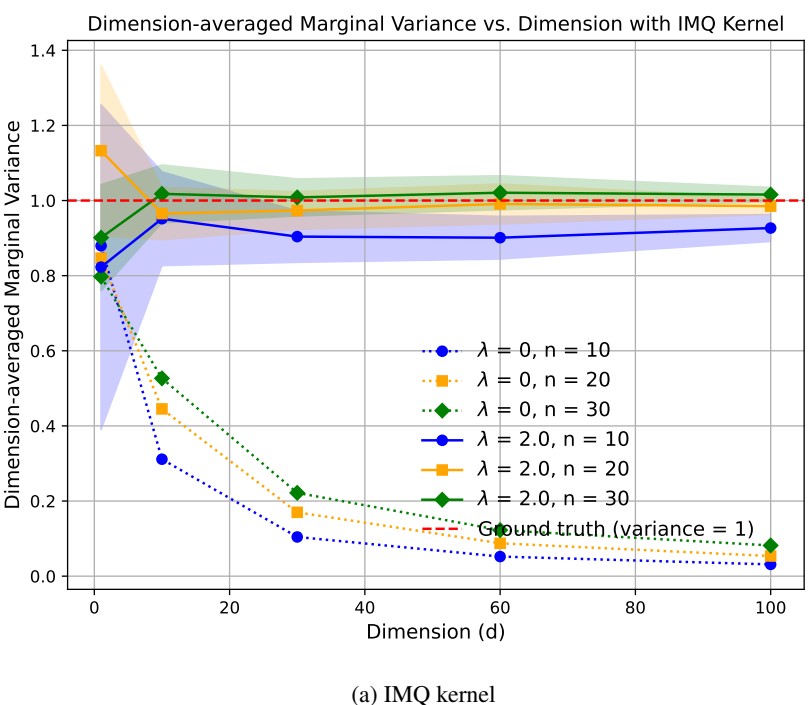

(a) IMQ kernel

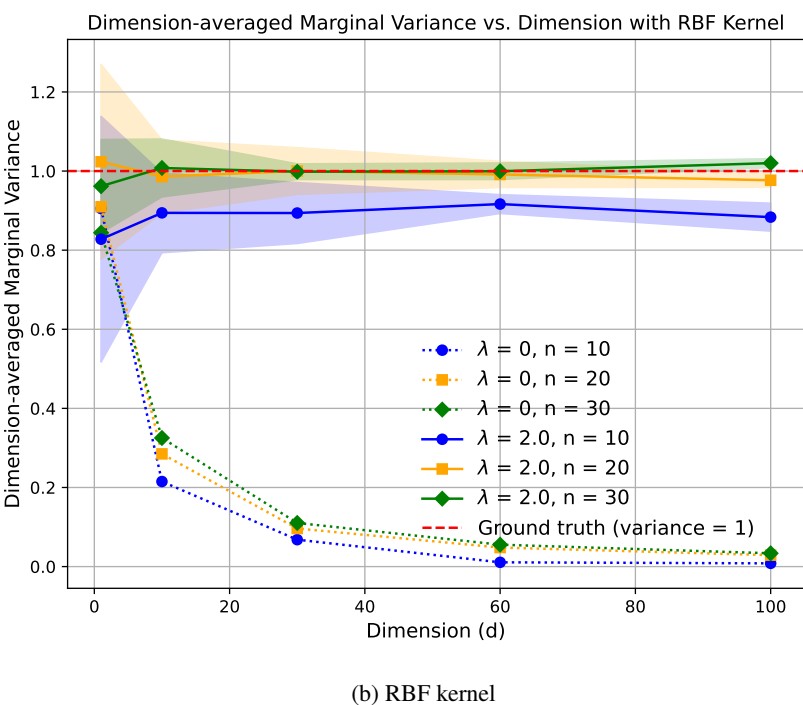

(b) RBF kernel

Figure 3: Dimension-averaged Marginal Variance of SVGD and NSVGD at convergence for sampling from a standard Gaussian.

## B  FIG. 2 IN LARGER FORMAT

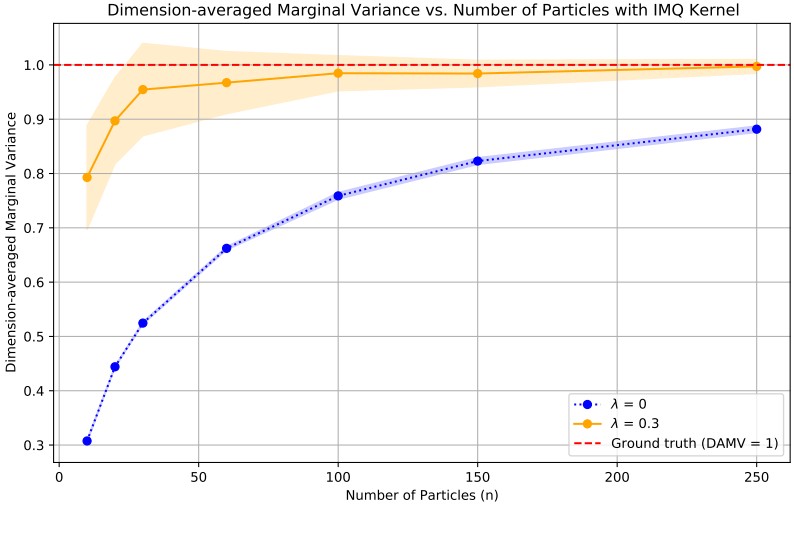

(a) IMQ kernel

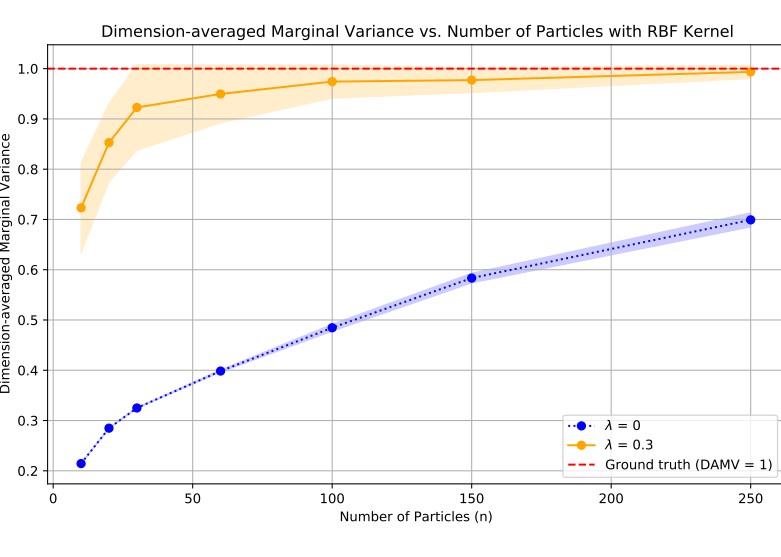

(b) RBF kernel

Figure 4: Dimension-averaged Marginal Variance of SVGD and NSVGD at convergence for sampling from a standard Gaussian for fixed dimension $d = 10$.

## C  ADDITIONAL EXPERIMENTS

In this section, we compare the Langevin algorithm (which is Eq. (2) with $\lambda = 1$ and $K = 0$) with NSVGD using various values of $\lambda$ and SVGD (which is NSVGD with $\lambda = 0$).

We consider the Neal funnel distribution (Neal, 2003), defined as:

$$\pi((x_1, x_2)) = \mathcal{N}(x_1; 0, 3)\mathcal{N}(x_2; 0, e^{x_1}),$$

where $x \mapsto \mathcal{N}(x; 0, \sigma^2)$ denotes the density of a centered Gaussian with variance $\sigma^2$.

We recall that the Maximum Mean Discrepancy (MMD) with a kernel $\tilde{K}$ between two distributions $\mu_1$ and $\mu_2$ is defined as:

$$\mathrm{MMD}^2(\mu_1, \mu_2) := \int \tilde{K}(x, x')\, d\mu_1(x)\, d\mu_1(x') + \int \tilde{K}(y, y')\, d\mu_2(y)\, d\mu_2(y')$$

$$- 2 \int \tilde{K}(x, y)\, d\mu_1(x)\, d\mu_2(y).$$

Given a sample of the target distribution ($N = 500$), denoted as $(X^1, \ldots, X^N)$, we plot the MMD distance between $\mu_k^n$ and $\frac{1}{N} \sum_{i \in [N]} \delta_{X^i}$ as a function of $k$ in Fig. 5.

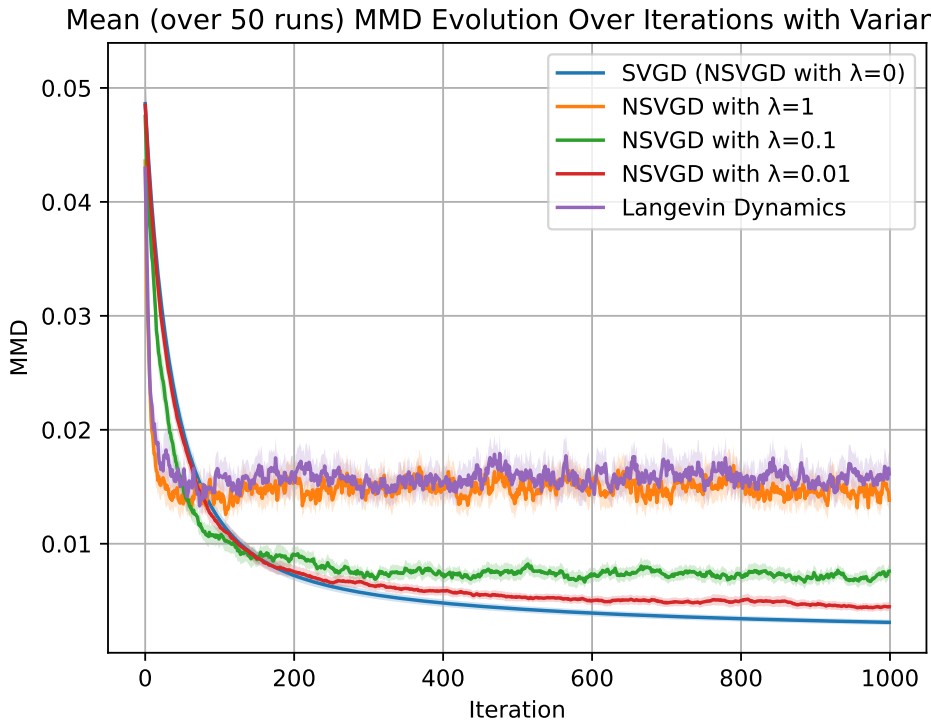

Figure 5: The RBF kernel is taken for both the MMD distance and the NSVGD algorithm. We consider $n = 100$ particles, $\gamma_k = 0.1$ and $d = 2$.

With this particular type of target distribution, the Langevin algorithm fails to recover the thin part of the distribution (see Fig. 6 when $x_1 \leq 2$), whereas SVGD successfully does so. This explains why SVGD performs better in this scenario. As described in Sec. 6, the performance of NSVGD lies between that of the Langevin and SVGD algorithms. By setting $\lambda$ small, NSVGD replicates the

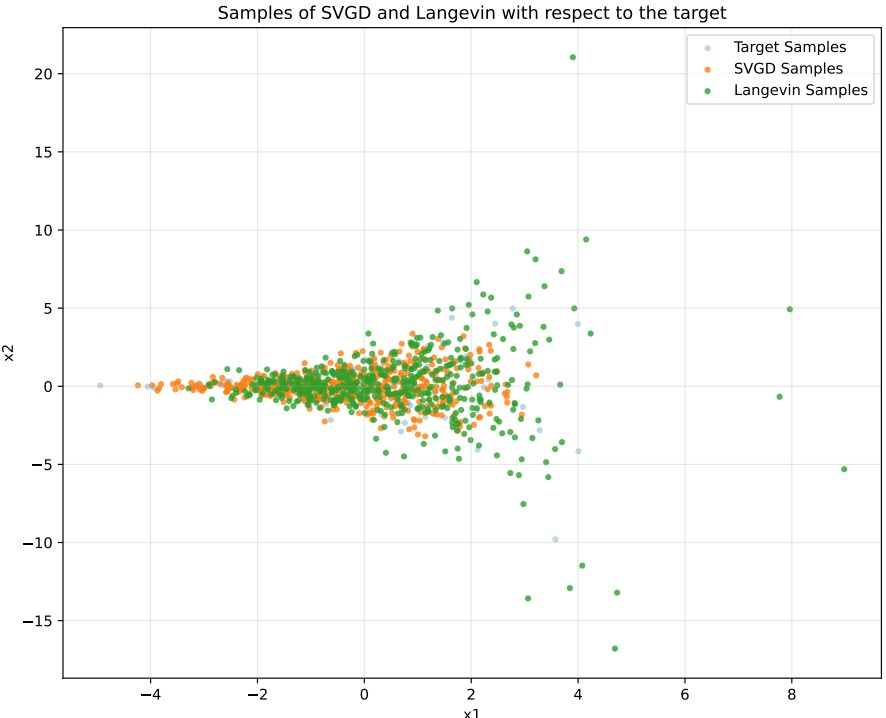

Figure 6: Plot of the particles $X_k^{i,n}$ after one run of the Langevin algorithm ($K = 0$ and $\lambda = 1$) and one run of the SVGD algorithm ($\lambda = 0$) with the RBF kernel and $n = 100$, $\gamma_k = 0.1$ $k = 500$.

performance of SVGD, while setting $\lambda$ large recovers the performance of the Langevin algorithm. The conclusion in this example is to use NSVGD with a small $\lambda$ to achieve similar performance to SVGD while maintaining convergence guarantees.

## D   ADDITIONAL NOTATIONS

In this section, we introduce further notations, extending what is provided in Sec. 2.

Let $d \in \mathbb{N}^*$. For $k \in \mathbb{N} \cup \{\infty\}$, we denote by $C^k(\mathbb{R}^d, \mathbb{R}^q)$ the set of functions which are continuously differentiable up to the order $k$. We denote by $C_c(\mathbb{R}^d, \mathbb{R})$ the set of $\mathbb{R}^d \to \mathbb{R}$ continuous functions with compact support. Given $p \in \mathbb{N}^* \cup \{\infty\}$, we denote as $C_c^p(\mathbb{R}^d, \mathbb{R})$ the set of compactly supported $\mathbb{R}^d \to \mathbb{R}$ functions which are continuously differentiable up to the order $p$.

The notation $f_{\#}\mu$ stands for the pushforward of the measure $\mu$ by the map $f$, that is, $f_{\#}\mu = \mu \circ f^{-1}$.

For $t \geq 0$, we define the projections $\pi_t$ and $\pi_{[0,t]}$ as $\pi_t : (\mathbb{R}^d)^{[0,\infty)} \to \mathbb{R}^d, x \mapsto x_t$, and $\pi_{[0,t]} : (\mathbb{R}^d)^{[0,\infty)} \to (\mathbb{R}^d)^{[0,t]}, x \mapsto (x_u : u \in [0,t])$.

Define:

$$\mathcal{P}_2(\mathcal{C}) = \{\rho \in \mathcal{P}(\mathcal{C}) : \forall T > 0, \int \sup_{t \in [0,T]} \|x_t\|^2 d\rho(x) < \infty\}.$$

For every $\rho, \rho' \in \mathcal{P}_2(\mathcal{C})$, we define:

$$\mathsf{W}_2(\rho, \rho') = \sum_{n=1}^{\infty} 2^{-n}(1 \wedge W_2((\pi_{[0,n]})_{\#}\rho, (\pi_{[0,n]})_{\#}\rho')),$$

where we equipped the space of the $[0,n] \to \mathbb{R}^d$ continuous function with the uniform norm for every $n \in \mathbb{N}^*$. We equip $\mathcal{P}_2(\mathcal{C})$ with the distance $\mathsf{W}_2$. By Bianchi et al. (2024, Prop. 1), $\mathcal{P}_2(\mathcal{C})$ is a Polish space.

For $\rho \in \mathcal{P}_2(\mathcal{C})$, we denote

$$\rho_t := (\pi_t)_{\#}\rho.$$

## E   PROOF OF PROP. 1

In this section, we let Assumptions 1 and 2 hold. Additionally, we assume $\lambda > 0$. Furthermore, $C > 0$ will denote a generic and sufficiently large constant independent of $k$ and $n$.

We define:

$$I_{k,n} := \frac{1}{n} \sum_{i \in [n]} F(X_k^{i,n}).$$

We will proceeds in three steps. First, we will obtain:

**Lemma 1.** *The following holds:*

$$\sup_{k,n} \mathbb{E}(I_{k,n}) < \infty.$$

Secondly:

**Lemma 2.** *The following holds:*

$$\sup_{k,n} \mathbb{E}(I_{k,n}^2) < \infty.$$

The latter lemma gives a bound on the cross terms of the form $\mathbb{E}(F(X_k^{i,n})F(X_k^{j,n}))$ for $i \neq j$. With this at hand, we obtain:

**Lemma 3.** *The following holds:*

$$\sup_{k,n} \mathbb{E}(F(X_k^{1,n})^2) < \infty.$$

$F(x) \geq c' \|x\|^2 - C$ by Assumtion 2. Hence, by Lem. 3, Prop. 1 is proven.

**Proof of Lem. 1** By Taylor-Lagrange formula, there exists $t_{k+1}^{i,n} \in [0,1]$ such that:

$$F(X_{k+1}^{i,n}) = F(X_k^{i,n}) + \langle \nabla F(X_k^{i,n}), X_{k+1}^{i,n} - X_k^{i,n} \rangle +$$
$$\frac{1}{2}\left(\left(X_{k+1}^{i,n} - X_k^{i,n}\right)^T \mathbf{H}(F)\left(X_{k+1}^{i,n} + t_{k+1}^{i,n}\left(X_{k+1}^{i,n} - X_k^{i,n}\right)\right)\left(X_{k+1}^{i,n} - X_k^{i,n}\right)\right). \quad (3)$$

We recall the iteration Eq. (2)

$$X_{k+1}^{i,n} - X_k^{i,n} = -\frac{\gamma_{k+1}}{n}\sum_{j\in[n]}\left(K(X_k^{i,n}, X_k^{j,n})\nabla F(X_k^{j,n}) - \nabla_y K(X_k^{i,n}, X_k^{j,n})\right)$$
$$- \lambda\gamma_{k+1}\nabla F(X_k^{i,n}) + \sqrt{2\gamma_{k+1}\lambda}\xi_{k+1}^{i,n}.$$

By Assumption 2, $\|\mathbf{H}(F)(x)\|_{op} \leq C$ for every $x \in \mathbb{R}^d$. Using Eq. (3), we obtain

$$F(X_{k+1}^{i,n}) \leq F(X_k^{i,n}) - \frac{\gamma_{k+1}}{n}\sum_{j\in[n]}\langle \nabla F(X_k^{i,n}), \nabla F(X_k^{j,n})\rangle K(X_k^{i,n}, X_k^{j,n})$$
$$+ \frac{\gamma_{k+1}}{n}\sum_{j\in[n]}\langle \nabla F(X_k^{i,n}), \nabla_y K(X_k^{i,n}, X_k^{j,n})\rangle + \sqrt{2\gamma_{k+1}\lambda}\langle \nabla F(X_k^{i,n}), \xi_{k+1}^{i,n}\rangle$$
$$+ C\gamma_{k+1}^2\left(\left\|\frac{1}{n}\sum_{j\in[n]}K(X_k^{i,n}, X_k^{j,n})\nabla F(X_k^{j,n})\right\|^2 + \left\|\frac{1}{n}\sum_{j\in[n]}\nabla_y K(X_k^{i,n}, X_k^{j,n})\right\|^2\right)$$
$$- \lambda\gamma_{k+1}\left\|\nabla F(X_k^{i,n})\right\|^2 + C\lambda^2\gamma_{k+1}^2\left\|\nabla F(X_k^{i,n})\right\|^2 + C\lambda\gamma_{k+1}\left\|\xi_{k+1}^{i,n}\right\|^2.$$

Note that

$$\frac{1}{n}\sum_{j\in[n]}\langle \nabla F(X_k^{i,n}), \nabla_y K(X_k^{i,n}, X_k^{j,n})\rangle \leq C\left\|\nabla F(X_k^{i,n})\right\|.$$

We remark that for an arbitrary $\Phi = (\Phi_\ell)_{\ell\in[d]} \in \mathcal{H}$, and for every $y \in \mathbb{R}^d$

$$\|\Phi(y)\|^2 = \sum_{\ell\in[d]}\langle \Phi_\ell, K(\cdot, y)\rangle_{\mathcal{H}_0}^2 \leq \sum_{\ell\in[d]}\|\Phi_\ell\|_{\mathcal{H}_0}^2\|K(\cdot, y)\|_{\mathcal{H}_0}^2 \leq C\|\Phi\|_{\mathcal{H}}^2.$$

Therefore,

$$\left\|\nabla_y K(X_k^{i,n}, X_k^{j,n})\right\|^2 \leq C\left\|\nabla_y K(\cdot, X_k^{j,n})\right\|_{\mathcal{H}}^2 \leq C,$$

and

$$\left\|\sum_{j\in[n]}K(X_k^{i,n}, X_k^{j,n})\nabla F(X_k^{j,n})\right\|^2 \leq C\left\|\sum_{j\in[n]}K(\cdot, X_k^{j,n})\nabla F(X_k^{j,n})\right\|_{\mathcal{H}}^2.$$

Consequently, we obtain

$$F(X_{k+1}^{i,n}) \leq F(X_k^{i,n}) - \frac{\gamma_{k+1}}{n}\sum_{j\in[n]}\langle \nabla F(X_k^{i,n}), \nabla F(X_k^{j,n})\rangle K(X_k^{i,n}, X_k^{j,n})$$
$$+ \gamma_{k+1}C\left\|\nabla F(X_k^{i,n})\right\| + \sqrt{2\gamma_{k+1}\lambda}\langle \nabla F(X_k^{i,n}), \xi_{k+1}^{i,n}\rangle$$
$$+ C\gamma_{k+1}^2\left(\left\|\frac{1}{n}\sum_{j\in[n]}K(\cdot, X_k^{j,n})\nabla F(X_k^{j,n})\right\|_{\mathcal{H}}^2 + 1\right)$$
$$- \lambda\gamma_{k+1}(1 - C\lambda\gamma_{k+1})\left\|\nabla F(X_k^{i,n})\right\|^2 + C\lambda\gamma_{k+1}\left\|\xi_{k+1}^{i,n}\right\|^2. \quad (4)$$

We define $J_{k,n} := \frac{1}{n} \sum_{i \in [n]} \left\| \nabla F(X_k^{i,n}) \right\|^2$. Hence, we obtain

$$I_{k+1,n} \leq I_{k,n} - \gamma_{k+1}(1 - C\gamma_{k+1}) \left\| \frac{1}{n} \sum_{j \in [n]} K(\cdot, X_k^{j,n}) \nabla F(X_k^{j,n}) \right\|_{\mathcal{H}}^2$$
$$- \lambda\gamma_{k+1}(1 - C\lambda\gamma_{k+1})J_{k,n} + \gamma_{k+1}C\sqrt{J_{k,n}}$$
$$+ \sqrt{2\gamma_{k+1}\lambda}\frac{1}{n} \sum_{i \in [n]} \langle \nabla F(X_k^{i,n}), \xi_{k+1}^{i,n} \rangle + C\lambda\gamma_{k+1}\frac{1}{n} \sum_{i \in [n]} \left\| \xi_{k+1}^{i,n} \right\|^2 + C\gamma_{k+1}^2 .$$

By Assumption 2, $c'I_{k,n} - C \leq J_{k,n} \leq C'I_{k,n} + C$. Hence, for $k$ large enough, there exist a constant $c > 0$ small enough

$$I_{k+1,n} \leq I_{k,n}(1 - c\gamma_{k+1}) + C\gamma_{k+1}\sqrt{C'I_{k,n} + C}$$
$$+ \sqrt{2\gamma_{k+1}\lambda}\frac{1}{n} \sum_{i \in [n]} \langle \nabla F(X_k^{i,n}), \xi_{k+1}^{i,n} \rangle + C\lambda\gamma_{k+1}\frac{1}{n} \sum_{i \in [n]} \left\| \xi_{k+1}^{i,n} \right\|^2 + C\gamma_{k+1} . \quad (5)$$

Taking the expectation in Eq. (5), we obtain by Assumption 1:

$$\mathbb{E}\left[I_{k+1,n}\right] \leq \mathbb{E}\left[I_{k,n}\right](1 - c\gamma_{k+1}) + C\gamma_{k+1}\sqrt{C'\mathbb{E}\left[I_{k,n}\right] + C} + C\gamma_{k+1} .$$

There exists a constant $\kappa$ large enough satisfying

$$c\kappa \geq C\sqrt{C'\kappa + C} + C .$$

Hence, as soon as there exists $k$ large enough such that $\mathbb{E}\left[I_{k,n}\right] \geq \kappa$, we obtain $\mathbb{E}\left[I_{k+1,n}\right] \leq \mathbb{E}\left[I_{k,n}\right]$. Consequently, since $\kappa$ is independent of $n$, Lem. 1 is proven.

**Proof of Lem. 2**  Raising Eq. (5) to the square and taking the expectation, we obtain for $k$ large enough, the existence of a constant $\tilde{c} > 0$ small enough, such that

$$\mathbb{E}\left[I_{k+1,n}^2\right] \leq \mathbb{E}\left[I_{k,n}^2\right](1 - \tilde{c}\gamma_{k+1}) + C\gamma_{k+1}\mathbb{E}\left[I_{k,n}^2\right]^{3/4} + C\gamma_{k+1}\mathbb{E}\left[I_{k,n}^2\right]^{1/2} + C\gamma_{k+1}^2 .$$

As in the proof of Lem. 1, Lem. 2 is proven.

**Proof of Lem. 3**  By Assumption 1, the sequence $(X_k^{i,n})_{i \in [n]}$ is exchangeable, i.e. the sequence is invariant in law by permutation of the indices $i \in [n]$. Then, by Lem. 2, we obtain

$$\sup_{k,n} \left( \frac{n-1}{n}\mathbb{E}\left[F(X_k^{1,n})F(X_k^{2,n})\right] + \frac{1}{n}\mathbb{E}\left[F(X_k^{1,n})^2\right] \right) < \infty . \quad (6)$$

Going back to Eq. (4) and raising it to the square and taking the expectation, using $\left\| \nabla F(x) \right\|^2 \leq C(|F(x)| + 1)$ and the exchangeability of $(X_i^{k,n})_{i \in [n]}$, we obtain the existence of a constant $\tilde{c}$ small enough, such that

$$\mathbb{E}\left[F(X_{k+1}^{1,n})^2\right] \leq \mathbb{E}\left[F(X_k^{1,n})^2\right](1 - \tilde{c}\gamma_{k+1})$$
$$+ C\gamma_{k+1}\left( \frac{n-1}{n}\mathbb{E}\left| \langle \nabla F(X_k^{1,n}), \nabla F(X_k^{2,n})\rangle F(X_k^{1,n}) \right| + \frac{1}{n}\mathbb{E}\left[ \left\| \nabla F(X_k^{1,n}) \right\|^2 \left| F(X_k^{1,n}) \right| \right] \right)$$
$$+ C\gamma_{k+1}\mathbb{E}\left[ \left\| \nabla F(X_k^{1,n}) \right\| \left| F(X_k^{1,n}) \right| \right] + C\gamma_{k+1}\mathbb{E}\left[ \left| F(X_k^{i,n}) \right| \right] + C\gamma_{k+1} .$$
$$(7)$$

In the above inequality, we didn't write the terms in $\gamma_k^2$ as they are dominated by the terms in $\gamma_k$. In the rest of the proof, we bound the second term on the right-hand side of the above inequality. The other terms are easier and are left to the reader. By Cauchy-Schwarz inequality, we obtain

$$\mathbb{E}\left[ \langle \nabla F(X_k^{1,n}), \nabla F(X_k^{2,n})\rangle F(X_k^{1,n}) \right] \leq \sqrt{\mathbb{E}\left[F(X_k^{1,n})^2\right]} \sqrt{\mathbb{E}\left[ \left\| \nabla F(X_k^{1,n}) \right\|^2 \left\| \nabla F(X_k^{2,n}) \right\|^2 \right]} .$$

Moreover, by Assumption 2, $\|\nabla F(x)\|^2 \leq C'F(x) + C$, and

$$\left\|\nabla F(X_k^{1,n})\right\|^2 \left\|\nabla F(X_k^{2,n})\right\|^2 \leq C'^2 F(X_k^{1,n})F(X_k^{2,n}) + CC'F(X_k^{1,n}) + C'CF(X_k^{2,n}) + C^2 \,.$$

By Eq. (6),

$$\mathbb{E}\left[\left\|\nabla F(X_k^{1,n})\right\|^2 \left\|\nabla F(X_k^{2,n})\right\|^2\right] \leq C(1 + \sqrt{\mathbb{E}\left[F(X_k^{1,n})^2\right]}) \,.$$

Hence, we obtain

$$\mathbb{E}\left|\langle\nabla F(X_k^{1,n}), \nabla F(X_k^{2,n})\rangle F(X_k^{1,n})\right| \leq C\left(\mathbb{E}\left[F(X_k^{1,n})^2\right]^{1/2} + \mathbb{E}\left[F(X_k^{1,n})^2\right]^{3/4}\right) \,.$$

By Eq. (6), we also obtain

$$\frac{1}{n}\mathbb{E}\left[\left\|\nabla F(X_k^{1,n})\right\|^2 \left|F(X_k^{1,n})\right|\right] \leq \frac{C}{n}(\mathbb{E}\left[F(X_k^{1,n})^2\right] + \mathbb{E}\left|F(X_k^{1,n})\right|) \leq C \,.$$

Going back to Eq. (7), we obtain

$$\mathbb{E}\left[F(X_{k+1}^{1,n})^2\right] \leq \mathbb{E}\left[F(X_k^{1,n})^2\right](1 - \tilde{c}\gamma_{k+1}) + C\gamma_{k+1}(\mathbb{E}\left[F(X_k^{1,n})^2\right]^{\frac{1}{2}} + \mathbb{E}\left[F(X_k^{1,n})^2\right]^{\frac{3}{4}} + 1) \,.$$

Hence, $\sup_{k,n}\mathbb{E}\left[F(X_k^{1,n})^2\right] < \infty$.

## F  TIGHTNESS RESULTS

We define the *intensity* of a random variable $\nu : \Omega \to \mathcal{P}_2(\mathbb{R}^d)$, as the measure $\mathbb{I}(\nu) \in \mathcal{P}(\mathbb{R}^d)$ that satisfies

$$\forall A \in \mathcal{B}(\mathbb{R}^d), \quad \mathbb{I}(\nu)(A) := \mathbb{E}\left(\nu(A)\right) \,.$$

**Lemma 4.** *A sequence $(\mu_n)$ of random variables on $\mathcal{P}_2(\mathbb{R}^d)$ is tight if the sequence $(\mathbb{I}(\mu_n))$ is relatively compact in $\mathcal{P}_2(\mathbb{R}^d)$.*

*Proof.* This proof is identical to the one presented in Bianchi et al. (2024, Lem. 2). □

### F.1  PROOF OF TH. 1 AND PROP. 2

First, we state a more general result, which is a consequence of Prop. 1.

**Lemma 5.** *Bianchi et al. (2024, Prop. 4) The collection of measure $(\mathbb{I}(m_t^n))_{t,n}$ is relatively compact in $\mathcal{P}_2(\mathcal{C})$. Moreover, the collection of random variables $(m_t^n)_{t,n}$ is tight.*

Next, as the consequence of the above lemma, we obtain the proof of Prop. 2.

**Proof of Prop. 2**  This is given by Bianchi et al. (2024, Lem. 8).

**Proof of Th. 1**  Remark that $(\pi_0)_{\#}m_{\tau_k}^n = \mu_k^n$, for every $k$. Hence, $(\pi_0)_{\#}\mathbb{I}(m_{\tau_k}^n) = \mathbb{I}(\mu_k^n)$. For a compact set $\mathcal{K} \subset \mathcal{P}_2(\mathcal{C})$, one can obtain that $(\pi_0)_{\#}\mathcal{K}$ is a compact set in $\mathcal{P}_2(\mathbb{R}^d)$. Consequently, since $\mathbb{I}(m_t^n)_{t,n}$ is relatively compact in $\mathcal{P}_2(\mathcal{C})$ by Lem. 5, $(\mathbb{I}(\mu_k^n))_{k,n}$ is relatively compact in $\mathcal{P}_2(\mathbb{R}^d)$. This yields the first claim of the theorem, by Lem. 4.

Moreover,

$$\mathbb{I}(\bar{\mu}_k^n) = \frac{\sum_{i\in[k]}\gamma_i\mathbb{I}(\mu_i^n)}{\sum_{i\in[k]}\gamma_i} \,.$$

Since, $(\mathbb{I}(\mu_k^n))_{k,n}$ is relatively compact in $\mathcal{P}_2(\mathbb{R}^d)$, the same holds for $(\mathbb{I}(\bar{\mu}_k^n))_{k,n}$. The proof is left to the reader. By Lem. 4, this finishes the proof.

## G  THE MCKEAN-VLASOV MEASURES

For every $\mu \in \mathcal{P}_2(\mathbb{R}^d)$, we define $L(\mu)$ which, to every test function $\phi \in C_c^2(\mathbb{R}^d, \mathbb{R})$, associates the function $L(\mu)(\phi)$ given by

$$L(\mu)(\phi)(x) = \langle \int (-K(x,y)\nabla F(y) + \nabla_y K(x,y))d\mu(y) - \lambda \nabla F(x), \nabla \phi(x)\rangle + \lambda \Delta \phi(x). \quad (8)$$

Let $(X_t : t \in [0,\infty))$ be the canonical process on $\mathcal{C}$. Denote by $(\mathcal{F}_t^X)_{t\geq 0}$ the natural filtration (*i.e.*, the filtration generated by $\{X_s : 0 \leq s \leq t\}$).

By a weak solution of the McKean-Vlasov SDE in Definition 4, we mean a solution of the martingale problem defined hereafter. Hence, for the rest of the appendix, we will take the subsequent definition of $\mathbb{V}_2$ into account.

*Definition* 5. We say that a measure $\rho \in \mathcal{P}_2(\mathcal{C})$ belongs to the class $\mathbb{V}_2$ if, for every $\phi \in C_c^2(\mathbb{R}^d, \mathbb{R})$,

$$\phi(X_t) - \int_0^t L(\rho_s)(\phi)(X_s)ds$$

is a $(\mathcal{F}_t^X)_{t\geq 0}$-martingale on the probability space $(\mathcal{C}, \mathcal{B}(\mathcal{C}), \rho)$.

We define the function

$$b(x,y) := -K(x,y)\nabla F(y) + \nabla_y K(x,y) - \lambda \nabla F(x)$$

With a slight abuse of notation, for a measure $\mu \in \mathcal{P}(\mathbb{R}^d)$, we denote $b(x,\mu) := \int b(x,y)d\mu(y)$. Therefore, $L(\mu)(\phi)(x) = \langle b(x,\mu), \nabla \phi(x)\rangle + \lambda \Delta \phi(x)$. When $b$ is continuous with linear growth, i.e. $\|b(x,y)\| \leq C(1 + \|x\| + \|y\|)$ for every $x, y \in \mathbb{R}^d$, the space $\mathbb{V}_2$ is Polish.

**Lemma 6.** *Bianchi et al. (2024, Prop. 3) Let Assumption 2 holds. $\mathbb{V}_2$ is closed. Consequently, the space $(\mathbb{V}_2, \mathsf{W}_2)$ is Polish.*

In the rest of the appendix, we will use the following property to derive results about the space $\mathbb{V}_2$.

**Proposition 6.** *Let $\rho \in \mathbb{V}_2$. Let Assumption 2 holds. Let $\psi \in C_c^\infty(\mathbb{R}_+ \times \mathbb{R}^d)$, then for every $t_2 \geq t_1 \geq 0$, we obtain*

$$\int \psi(t_2, x)d\rho_{t_2}(x) - \int \psi(t_1, x)d\rho_{t_1}(x) = \int_{t_1}^{t_2}\int \partial_t \psi(t,x)d\rho_t(x)dt$$

$$+ \int_{t_1}^{t_2}\int \langle \nabla \psi(t,x), b(x,\rho_t)\rangle d\rho_t(x)dt + \lambda \int_{t_1}^{t_2}\int \Delta \psi(t,x)d\rho_t(x)dt. \quad (9)$$

*Proof.* Let $\phi \in C_c^\infty(\mathbb{R}^d)$. Let $\rho \in \mathbb{V}_2$. By Def. 5, the function

$$t \in \mathbb{R}_+ \mapsto \int \phi(x)d\rho_t(x) - \int_0^t \int L(\rho_s)(\phi)(x)d\rho_s(x)ds$$

is constant. Hence, the function $\Phi(t) := \int \phi(x)d\rho_t(x)$ is absolutely continuous, with derivative $\Phi'(t) = \int L(\rho_t)(\phi)(x)d\rho_t(x)$, which is bounded on compacts under Assumption 2. Let $\eta \in C_c^\infty(\mathbb{R}_+)$, by an integration by parts, we obtain for every $t_2 > t_1 \geq 0$

$$\Phi(t_2)\eta(t_2) - \Phi(t_1)\eta(t_1) = \int_{t_1}^{t_2} \Phi'(t)\eta(t) + \Phi(t)\eta'(t)dt.$$

Hence, if we define $\psi(t,x) := \phi(x)\eta(t)$, we obtain Eq. (9). It suffices to remark that functions of the form $(t,x) \mapsto \phi(x)\eta(t)$ for every $(\eta, \phi) \in C_c^\infty(\mathbb{R}_+) \times C_c^\infty(\mathbb{R}^d)$ are dense in $C_c^\infty(\mathbb{R}_+ \times \mathbb{R}^d)$, and the proof is finished.  $\square$

**Lemma 7.** *Let Assumptions 2 and 3 hold. Moreover, we assume $\lambda > 0$. Let $\rho \in \mathbb{V}_2$. For every $t > 0$, $\rho_t$ admits a density $x \mapsto \varrho(t, x) \in C^1(\mathbb{R}^d, \mathbb{R})$. Moreover, for every $R > 0, t_2 > t_1 > 0$, there exists a constant $C_{R,t_1,t_2} > 0$ such that:*

$$\inf_{t\in[t_1,t_2], \|x\|\leq R} \varrho(t, x) \geq C_{R,t_1,t_2}, \quad (10)$$

*and there exist a constant $C_{t_1,t_2} > 0$, such that*

$$\sup_{x\in\mathbb{R}^d,t\in[t_1,t_2]} \|\nabla\varrho(t,x)\| + \varrho(t,x) \leq C_{t_1,t_2}. \tag{11}$$

*Additionally,*

$$\sup_{t\in[t_1,t_2]} \int (1 + \|x\|^2) \|\nabla\varrho(t,x)\| \, dx < \infty. \tag{12}$$

*Finally,*

$$\sup_{\rho\in\mathcal{K}} D_{\mathrm{KL}}(\rho_{t_1}\|\pi) < \infty, \tag{13}$$

*for every compact set $\mathcal{K} \subset \mathbb{V}_2$.*

*Proof.* The result is an application of Menozzi et al. (2021, Th. 1.2) with the non homogeneous vector field $\tilde{b}(t,x) := \int b(x,y)d\rho_t(y)$. The proof consists in verifying the conditions of the latter theorem. By Assumptions 2 and 3, for every $(x,y,T) \in (\mathbb{R}^d)^2 \times \mathbb{R}_+$,

$$\sup_{t\in[0,T]} \left\|\tilde{b}(t,x) - \tilde{b}(t,y)\right\| \leq \lambda \|\nabla F(x) - \nabla F(y)\|$$

$$+ \sup_{t\in[0,T]} \int \|\nabla_y K(x,z) - \nabla_y K(y,z)\| \, d\rho_t(z)$$

$$+ \sup_{t\in[0,T]} \int \|\nabla F(z)\| \, |K(x,z) - K(y,z)| \, d\rho_t(z)$$

$$\leq C(\|x-y\|^\beta \vee \|x-y\|),$$

Moreover,

$$\sup_{t\in[0,T]} \tilde{b}(t,x) \leq C(1 + \|x\| + \int \sup_{t\in[0,T]} \|y_t\| \, d\rho(y)) \leq C(1 + \|x\|). \tag{14}$$

As $\lambda > 0$, Menozzi et al. (2021, Th. 1.2) applies: $\rho$ admits a density $x \mapsto \varrho(t,x) \in C^1(\mathbb{R}^d)$, for $0 < t \leq T$, and there exists four constants $(C_{i,T}, \lambda_{i,T})_{i\in[2]}$, such that:

$$\frac{1}{C_{1,T}t^{d/2}} \int \exp\left(-\frac{\|x - \theta_t(y)\|^2}{\lambda_{1,T}t}\right) d\rho_0(y) \leq \varrho(t,x)$$

$$\varrho(t,x) \leq \frac{C_{1,T}}{t^{d/2}} \int \exp\left(-\frac{\lambda_{1,T}}{t} \|x - \theta_t(y)\|^2\right) d\rho_0(y)$$

$$\|\nabla\varrho(t,x)\| \leq \frac{C_{2,T}}{t^{(d+1)/2}} \int \exp\left(-\frac{\lambda_{2,T}}{t} \|x - \theta_t(y)\|^2\right) d\rho_0(y),$$

where the map $t \mapsto \theta_t(y)$ is a solution to the ordinary differential equation: $\frac{d\theta_t(y)}{dt} = \tilde{b}(t,\theta_t(y))$ with initial condition $\theta_0(y) = y$. By Grönwall's lemma and Eq. (14), there exists a constant $C_T$ such that $\|\theta_t(y)\| \leq C_T\|y\|$, for every $n, y$, and $t \leq T$. For every $t_1 \leq t \leq t_2$, and every $x$, we obtain using a change of variables:

$$(C_{1,t_2}t_1^{d/2})^{-1} \geq \varrho(t,x) \geq C_{1,t_2}t_2^{-d/2} \exp\left(-\frac{2}{\lambda_{1,t_2}t_1}\|x\|^2\right) \int \exp\left(-\frac{2C_{t_2}}{\lambda_{1,t_2}t_1}\|y\|^2\right) d\rho_0(y)$$

$$\int (1 + \|x\|^2) \|\nabla\varrho(t,x)\| \, dx$$

$$\leq C_{2,t_2}t_1^{-(d+1)/2} \int (1 + 2\|x\|^2 + 2C_{t_2}^2 \int \|y\|^2 d\rho_0(y)) \exp\left(-\lambda_{2,t_2}t_2^{-1}\|x\|^2\right) dx,$$

and $\|\nabla\varrho(t,x)\| \leq C_{2,t_2}t_1^{-(d+1)/2}$. Consequently, $\rho$ satisfies Eq. (10), Eq. (11) and Eq. (12).

It remains to obtain Eq. (13). Let $\mathcal{K} \subset \mathbb{V}_2$ be a compact set and let $\rho \in \mathcal{K}$. We observe

$$D_{\mathrm{KL}}(\rho_{t_1}\|\pi) \leq C + \int |F(x)| \, d\rho_{t_1}(x) + \int \|\log\varrho(t_1,x)\| \, d\rho_{t_1}(x). \tag{15}$$

By Assumption 2, since $(\pi_{t_1})_{\#}\mathcal{K}$ is a compact set in $\mathcal{P}_2(\mathbb{R}^d)$, we obtain

$$\sup_{\rho \in \mathcal{K}} \int |F(x)|\, d\rho_{t_1}(x) \leq C \sup_{\rho \in \mathcal{K}} \int \|x\|^2\, d\rho_{t_1}(x) \leq C \sup_{\mu \in (\pi_{t_1})_{\#}\mathcal{K}} \int \|x\|^2\, d\mu(x) < \infty.$$

Moreover, by the lower bound and the upper bound on $\varrho$,

$$\|\log \varrho(t_1, x)\| \leq C \left( 1 + \|x\|^2 + \int \|y\|^2\, d\rho_0(y) \right). \tag{16}$$

Hence, we obtain

$$\sup_{\rho \in \mathcal{K}} \int \|\log \varrho(t_1, x)\|\, d\rho_{t_1}(x) < \infty.$$

Finally, applying the latter results in Eq. (15), we obtain Eq. (13). $\qquad\square$

### G.1 Sketch of the proof of Prop 4 using Wasserstein calculus

We give a sketch of the proof of Lyapunov using Wasserstein calculus (Ambrosio et al., 2008). This proof is not fully rigorous because we would need to check the assumptions of the results from Ambrosio et al. (2008) that we are using. In the next section we give a fully rigorous proof.

In this subsection, $\langle \cdot, \cdot \rangle_\rho$ (resp. $\|\cdot\|_\rho$) denotes the standard inner product (resp. the norm) in $L^2(\rho)$.

Consider $\rho \in \mathbb{V}_2$, *i.e.*, the law of a weak solution $(X_t)_t$ of the McKean-Vlasov equation

$$dX_t = -\int \left( K(X_t, y)\nabla F(y) - \nabla_y K(X_t, y) \right) d\rho_t(y)\, dt - \lambda \nabla F(X_t)\, dt + \sqrt{2\lambda}\, dW_t.$$

For every $t > 0$, we denote by $\rho_t$ the marginal of $\rho$. In other words, $\rho_t$ is the law of $X_t$.

Using integration by parts, the McKean-Vlasov equation can be represented by

$$dX_t = -P_\mu \nabla \log \frac{d\rho_t}{d\pi}(X_t)\, dt - \lambda \nabla F(X_t)\, dt + \sqrt{2\lambda}\, dW_t.$$

From this representation, we can derive the continuity equation satisfied by $(\rho_t)_t$:

$$\frac{\partial \rho_t}{\partial t} + \nabla \cdot (\rho_t \tilde{v}_t) = 0,$$

where $\tilde{v}_t$ is the velocity field

$$\tilde{v}_t := -P_\mu \nabla \log \frac{d\rho_t}{d\pi} - \lambda \nabla \log \frac{d\rho_t}{d\pi}.$$

Using the chain rule in the Wasserstein space (Ambrosio et al., 2008, Equation 10.1.16), we have for every functional $\mathcal{F} : \mathcal{P}_2(\mathbb{R}^d) \to (-\infty, +\infty]$ regular enough that

$$\frac{d}{dt}\mathcal{F}(\rho_t) = \langle \nabla_W \mathcal{F}(\rho_t), v_t \rangle_{\rho_t},$$

where $\nabla_W \mathcal{F}(\rho) \in L^2(\rho)$ is the Wasserstein gradient of $\mathcal{F}$ at $\rho$. In the case where $\mathcal{F}(\rho) = D_{\mathrm{KL}}(\rho\|\pi)$, we have $\nabla_W \mathcal{F}(\rho) = \nabla \log \frac{d\rho}{d\pi}$, therefore

$$\frac{d}{dt}\mathcal{F}(\rho_t) = \left\langle \nabla \log \frac{d\rho}{d\pi}, -P_\mu \nabla \log \frac{d\rho_t}{d\pi} - \lambda \nabla \log \frac{d\rho_t}{d\pi} \right\rangle_{\rho_t}$$

$$= -\left\langle \nabla \log \frac{d\rho}{d\pi}, P_\mu \nabla \log \frac{d\rho_t}{d\pi} \right\rangle_{\rho_t} - \lambda \left\langle \nabla \log \frac{d\rho}{d\pi}, \nabla \log \frac{d\rho_t}{d\pi} \right\rangle_{\rho_t}.$$

Finally, we use that the kernel integral operator is the adjoint of the injection (Carmeli et al., 2010) $\iota_\rho : \mathcal{H} \to L^2(\rho)$. In other words, for every $f \in L^2(\rho), g \in \mathcal{H}, \langle f, g \rangle_\rho = \langle P_\rho f, g \rangle_\mathcal{H}$. Here, this property gives

$$\left\langle \nabla \log \frac{d\rho}{d\pi}, P_\mu \nabla \log \frac{d\rho_t}{d\pi} \right\rangle_{\rho_t} = \left\| P_\mu \nabla \log \frac{d\mu}{d\pi} \right\|_\mathcal{H}^2.$$

Therefore,

$$\frac{d}{dt}\mathcal{F}(\rho_t) = -\left\|P_\mu \nabla \log \frac{d\mu}{d\pi}\right\|_\mathcal{H}^2 - \lambda \left\|\nabla \log \frac{d\mu}{d\pi}\right\|_{\rho_t}^2.$$

In other words,

$$\frac{d}{dt}D_{\mathrm{KL}}(\rho_t||\pi) = -\mathcal{I}_{\mathrm{stein}}(\rho_t||\pi) - \lambda\mathcal{I}(\rho_t||\pi),$$

and we can conclude by integrating between $t_1 > 0$ and $t_2 > 0$.

### G.2 PROOF OF PROP. 4

In this subsection, we let Assumptions 2 and 3 hold. Moreover, we assume $\lambda > 0$.

We consider $\rho \in \mathbb{V}_2$. Moreover, we define two reels $0 < t_1 < t_2$.

Let

$$v_t(x) := -\int \left(K(x,y)\nabla F(y) - \nabla_y K(x,y)d\rho_t(y)\right) - \lambda\nabla F(x) - \lambda\nabla \log \varrho(t,x). \quad (17)$$

By Prop 6, with Lem. 7, we obtain

$$\int \psi(t_2, x)d\rho_{t_2}(x) - \int \psi(t_1, x)d\rho_{t_1}(x)$$
$$= \int_{t_1}^{t_2}\int \partial_t\psi(t,x)d\rho_t(x)dt + \int_{t_1}^{t_2}\int \langle\nabla\psi(t,x), v_t(x)\rangle d\rho_t(x)dt. \quad (18)$$

Note that the latter quantity is well-defined, since $\int_{t_1}^{t_2}\int \|v_t(x)\| d\rho_t(x)dt$ by Lem. 7. Define a smooth, compactly supported, even function $\eta : \mathbb{R}^d \to \mathbb{R}_+$ such that $\int \eta(x)dx = 1$, and define $\eta_\varepsilon(x) := \varepsilon^{-d}\eta(x/\varepsilon)$ for every $\varepsilon > 0$. For every $t > 0$, we introduce the density $\varrho_\varepsilon(t, \cdot) := \eta_\varepsilon * \rho_\varepsilon(t, \cdot)$, and we denote by $\rho_t^\varepsilon(dx) = \varrho_\varepsilon(t,x)dx$ the corresponding probability measure. Finally, we define:

$$v_t^\varepsilon := \frac{\eta_\varepsilon * (v_t\varrho(t,\cdot))}{\varrho_\varepsilon(t,\cdot)}.$$

With these definitions at hand, it is straightforward to check that Eq. (18) holds when $\rho_t, v_t$ are replaced by $\rho_t^\varepsilon, v_t^\varepsilon$. More specifically, we shall apply Eq. (18) using a specific smooth function $\psi = \psi_{\varepsilon,\delta,R}$, which we will define hereafter for fixed values of $\delta, R > 0$, yielding our main equation:

$$\int \psi_{\varepsilon,\delta,R}(t_2, x)\varrho_\varepsilon(t_2,x)dx - \int \psi_{\varepsilon,\delta,R}(t_1, x)\varrho_\varepsilon(t_1,x)dx =$$
$$\int_{t_1}^{t_2}\int \left(\partial_t\psi_{\varepsilon,\delta,R}(t,x) + \langle\nabla\psi_{\varepsilon,\delta,R}(t,x), v_t^\varepsilon(x)\rangle\right)\varrho_\varepsilon(t,x)dxdt. \quad (19)$$

Let $\theta \in C_c^\infty(\mathbb{R}, \mathbb{R})$ be a nonnegative function supported by the interval $[-t_1, t_1]$ and satisfying $\int \theta(t)dt = 1$. For every $\delta \in (0, 1)$, define $\theta_\delta(t) = \theta(t/\delta)/\delta$. We define $\varrho_{\varepsilon,\delta}(\cdot, x) := \theta_\delta * \varrho_\varepsilon(\cdot, x)$. The map $t \mapsto \varrho_{\varepsilon,\delta}(t, \grave{)}$ is well-defined on $[t_1, t_2]$, non negative, and smooth in both variables $t, x$. In addition, we define $F_\varepsilon := \eta_\varepsilon * F$. Finally, we introduce a smooth function $\chi$ on $\mathbb{R}^d$ equal to one on the unit ball and to zero outside the ball of radius 2, and we define $\chi_R(x) := \chi(x/R)$. For every $(t, x) \in [t_1, t_2] \times \mathbb{R}$, we define:

$$\psi_{\varepsilon,\delta,R}(t, x) := (\log \varrho_{\varepsilon,\delta}(t, x) + F_\varepsilon(x))\chi_R(x). \quad (20)$$

We extend $\psi_{\varepsilon,\delta,R}$ to a smooth compactly supported function on $\mathbb{R}_+ \times \mathbb{R}^d$. We define $U(x,\rho_t) := \int (K(x,y)\nabla F(y) - \nabla_y K(x,y)d\rho_t(y)$. Applying Eq. (19) with $\psi_{\varepsilon,\delta,R}$,

$$\int \psi_{\varepsilon,\delta,R}(t_2,x)d\rho_{t_2}(x) - \int \psi_{\varepsilon,\delta,R}(t_1,x)d\rho_{t_1}(x)$$

$$= \int_{t_1}^{t_2}\int (\partial_t\psi_{\varepsilon,\delta,R}(t,x) + \langle\nabla\psi_{\varepsilon,\delta,R}(t,x), v_t^\varepsilon(x)\rangle)d\rho_t^\varepsilon(x)dt$$

$$= \int_{t_1}^{t_2}\int \partial_t\varrho_{\varepsilon,\delta}(t,x)\frac{\varrho_\varepsilon(t,x)}{\varrho_{\varepsilon,\delta}(t,x)}\chi_R(x)dxdt$$

$$- \lambda\int_{t_1}^{t_2}\int \langle\nabla F_\varepsilon(x) + \nabla\log\varrho_{\varepsilon,\delta}(t,x), \frac{\eta_\epsilon * (\nabla F(\cdot)\varrho(t,\cdot))(x)}{\varrho_\varepsilon(t,x)} + \nabla\log\varrho^\varepsilon(t,x)\rangle\chi_R(x)d\rho_t^\varepsilon(x)dt$$

$$- \int_{t_1}^{t_2}\int \langle\nabla F_\varepsilon(x) + \nabla\log\varrho_{\varepsilon,\delta}(t,x), \frac{\eta_\epsilon * (U(\cdot,\rho_t)\varrho(t,\cdot))(x)}{\varrho_\varepsilon(t,x)}\rangle\chi_R(x)d\rho_t^\varepsilon(x)dt$$

$$+ \int_{t_1}^{t_2}\int (\log\varrho_{\varepsilon,\delta}(t,x) + F_\varepsilon(x))\langle\nabla\chi_R(x), v_t^\varepsilon(x)\rangle d\rho_t^\varepsilon(x)dt$$

We define, for every $t \in [t_1,t_2]$,

$$\Pi_1(t) := \int \psi_{\varepsilon,\delta,R}(t,x)d\rho_t^\varepsilon(x),$$

$$\Pi_2 := \int_{t_1}^{t_2}\int \partial_t\varrho_{\varepsilon,\delta}(t,x)\frac{\varrho_\varepsilon(t,x)}{\varrho_{\varepsilon,\delta}(t,x)}\chi_R(x)dxdt,$$

$$\Pi_3 := \int_{t_1}^{t_2}\int \langle\nabla F_\varepsilon(x) + \nabla\log\varrho_{\varepsilon,\delta}(t,x), \eta_\epsilon * (\nabla F(\cdot)\varrho(t,\cdot))(x) + \nabla\varrho^\varepsilon(t,x)\rangle\chi_R(x)dxdt,$$

$$\Pi_4 := \int_{t_1}^{t_2}\int \langle\nabla F_\varepsilon(x) + \nabla\log\varrho_{\varepsilon,\delta}(t,x), \eta_\epsilon * (U(\cdot,\rho_t)\varrho(t,\cdot))(x)\rangle\chi_R(x)dxdt,$$

$$\Pi_5 := \int_{t_1}^{t_2}\int (\log\varrho_{\varepsilon,\delta}(t,x) + F_\varepsilon(x))\langle\nabla\chi_R(x), v_t^\varepsilon(x)\rangle\varrho^\varepsilon(t,x)dxdt.$$

And, it holds:
$$\Pi_1(t_2) - \Pi_1(t_1) = \Pi_2 - \lambda\Pi_3 - \Pi_4 + \Pi_5. \tag{21}$$

We now investigate successively the limit of each term in Eq. (21) as $\delta, \varepsilon, R$ successively tend to $0, 0, \infty$.

We state a technical result proven at the end of the subsection.

**Lemma 8.** *For every $\varepsilon, x \in \mathbb{R}^d$, $t \mapsto \rho^\varepsilon(x,t)$ and $t \mapsto \nabla\varrho^\varepsilon(t,x)$ are absolute continuous functions. Moreover,*
$$\sup_{t\in[t_1,t_2], x\in\mathbb{R}^d}|\partial_t\varrho_\varepsilon(t,x)| \leq C_\varepsilon,$$

*for a constant $C_\varepsilon > 0$.*

Since, by Lem. 7, the mappings $t \mapsto \varrho_\varepsilon(t,x)$, $x \mapsto F(x)$ and $x \mapsto \varrho(t,x)$ are continuous, and by Eq (10), we obtain
$$\lim_{R\to\infty}\lim_{\varepsilon\to0}\lim_{\delta\to0}\psi_{\varepsilon,\delta,R}(t,x) = \log\varrho(t,x) + F(x). \tag{22}$$

By Lem. 7, we obtain
$$\psi_{\varepsilon,\delta,R}\varrho_\varepsilon(t,x) \leq C_R\chi_R(x),$$

for a constant $C_R$ independent of $\delta, \varepsilon, x$. Hence, we can apply the dominated convergence theorem and we obtain $\lim_{\varepsilon\to0}\lim_{\delta\to0}\Pi_1(t) = \int \log(\varrho(t,x) + F(x))\chi_R(x)d\rho_t(x)$. Since $\rho_t$ admits moments of order 2, we obtain

$$\lim_{R\to\infty}\lim_{\varepsilon\to0}\lim_{\delta\to0}\Pi_1(t) = D_{\mathrm{KL}}(\rho_t||\pi) - \int \exp(-F(x))dx,$$

for every $t > 0$.

In the following, we will obtain the convergence of $\Pi_2$. We obtain

$$\Pi_2 = \int_{t_1}^{t_2}\int \partial_t \varrho_{\varepsilon,\delta}(t,x)\chi_R(x)dxdt + \int_{t_1}^{t_2}\int \partial_t \varrho_{\varepsilon,\delta}(t,x)\left(\frac{\varrho_\varepsilon(t,x)}{\varrho_{\varepsilon,\delta}(t,x)} - 1\right)\chi_R(x)dxdt\,.$$

By Lem. 8, and a convergence dominated argument, we obtain

$$\lim_{\delta\to 0}\int_{t_1}^{t_2}\int \partial_t \varrho_{\varepsilon,\delta}(t,x)\left(\frac{\varrho_\varepsilon(t,x)}{\varrho_{\varepsilon,\delta}(t,x)} - 1\right)\chi_R(x)dxdt = 0\,.$$

Moreover,

$$\int_{t_1}^{t_2}\int \partial_t \varrho_{\varepsilon,\delta}(t,x)\chi_R(x)dxdt = \int \varrho_{\varepsilon,\delta}(t_2,x)\chi_R(x)dx - \int \varrho_{\varepsilon,\delta}(t_1,x)\chi_R(x)dx\,.$$

Since $\sup_{x\in\mathbb{R}^d, t>0}\varrho(t,x)\le C$, we obtain the by dominated convergence theorem

$$\lim_{R\to\infty}\lim_{\varepsilon\to 0}\lim_{\delta\to 0}\int \varrho_{\varepsilon,\delta}(t_2,x)\chi_R(x)dx - \int \varrho_{\varepsilon,\delta}(t_1,x)\chi_R(x)dx = \int d\rho_{t_2} - \int d\rho_{t_1} = 0\,.$$

Hence,

$$\lim_{R\to\infty}\lim_{\varepsilon\to 0}\lim_{\delta\to 0}\Pi_2 = 0\,.$$

Next, we will obtain the convergence of $\Pi_3$. By Lem. 7 and 8, we obtain

$$\lim_{\varepsilon\to 0}\lim_{\delta\to 0}\Pi_3 = \int_{t_1}^{t_2}\int \|\nabla F(x) + \nabla\log\varrho(t,x)\|^2\chi_R(x)\rho_t(x)dt\,.$$

And by the monotone convergence theorem, we obtain the limit in $R$:

$$\lim_{R\to\infty}\lim_{\varepsilon\to 0}\lim_{\delta\to 0}\Pi_3 = \int_{t_1}^{t_2}\int \|\nabla F(x) + \nabla\log\varrho(t,x)\|^2 d\rho_t(x)dt\,.$$

Now, we will obtain the convergence of $\Pi_4$. We recall that the kernel $K$ is bounded by Assumption 2. First, remark that an integration by parts yields,

$$U(x,\rho_t) = \int K(x,y)\left(\nabla F(y) + \nabla\log\varrho(t,y)\right)d\rho_t(y)\,,$$

for every $x\in\mathbb{R}^d$, which is possible by Lem. 7. Hence, taking the limit in $\delta,\varepsilon$, we obtain

$$\lim_{\varepsilon\to 0}\lim_{\delta\to 0}\Pi_4$$
$$= \int_{t_1}^{t_2}\iint K(x,y)\langle\nabla F(x) + \nabla\log\varrho(t,x), \nabla F(y) + \nabla\log\varrho(t,y)\rangle\chi_R(x)d\rho_t(x)d\rho_t(y)dt\,.$$

Since, by Lem. 7, $\sup_{t\in[t_1,t_2]}\int \|\nabla\varrho(t,x)\|\,dx < \infty$, we obtain

$$\sup_{t\in[t_1,t_2]}\int \|\nabla F(y) + \nabla\varrho(t,y)\|\,d\rho_t(y) < \infty\,.$$

Hence, taking the limit in $R$,

$$\lim_{R\to\infty}\lim_{\varepsilon\to 0}\lim_{\delta\to 0}\Pi_4$$
$$= \int_{t_1}^{t_2}\iint K(x,y)\langle\nabla F(x) + \nabla\log\varrho(t,x), \nabla F(y) + \nabla\log\varrho(t,y)\rangle d\rho_t(x)d\rho_t(y)dt\,.$$

It remains to study a last term: $\Pi_5$. And, we obtain by Lem. 7 and 8,

$$\lim_{\varepsilon\to 0}\lim_{\delta\to 0}\Pi_5 = \int_{t_1}^{t_2}\int (\log\varrho(t,x) + F(x))\langle\nabla\chi_R(x), v_t(x)\rangle d\rho_t(x)\,.$$

By Eq. (16) and (12),

$$\sup_{t\in[t_1,t_2]}\int \|(\log\varrho(t,x)+F(x))\nabla\varrho(t,x)\|\,dx<\infty\,.$$

Now, we remark that $\|\nabla\chi_R(x)\|\le\frac{C}{\|x\|}$. Then,

$$\sup_{t\in[t_1,t_2],x\in\mathbb{R}^d}\|\nabla\chi_R(x)\|\,\|U(x,\rho_t)+\nabla F(x)\|<\infty\,.$$

Consequently, by the two above equations, we can apply a dominated convergence theorem:

$$\lim_{R\to\infty}\lim_{\varepsilon\to0}\lim_{\delta\to0}\Pi_5=0\,.$$

Going back to Eq. (21), we have shown

$$D_{\mathrm{KL}}(\rho_{t_2}||\pi)-D_{\mathrm{KL}}(\rho_{t_1}||\pi)=-\int_{t_1}^{t_2}\mathcal{I}_{\mathrm{stein}}(\rho_t||\pi)+\lambda\mathcal{I}(\rho_t||\pi)dt\,.$$

**Proof of Lem. 8**    Using Eq. (19) and integration by parts,

$$\varrho^\varepsilon(t_2,x)-\varrho^\varepsilon(t_1,x)$$
$$=-\int_{t_1}^{t_2}\int\langle\nabla\eta_\varepsilon(x-y),b(y,\rho_s)\rangle d\rho_s(y)ds+\lambda\int_{t_1}^{t_2}\int\Delta\eta_\varepsilon(x-y)d\rho_s(y)ds\,.$$

Since $\rho\in\mathcal{P}_2(\mathcal{C})$, $\sup_{t\in[t_1,t_2]}\|b(y,\rho_t)\|\le C(1+\|y\|)+C\int\sup_{t\in[t_1,t_2]}\|x_t\|\,d\rho(x)$. As a consequence, $\sup_{t\in[1,T]}\|b(y,\rho_t)\|\le C(1+\|y\|)$. Along with the observation that, for any fixed $\varepsilon$, $\nabla\eta_\varepsilon$ and $\Delta\eta_\varepsilon$ are bounded, it follows that $t\mapsto\varrho^\varepsilon(t,x)$ is Lipschitz continuous on $[t_1,t_2]$, and that its derivative almost everywhere is given by: $\partial_t\varrho^\varepsilon(t,x)=\int(\langle\nabla\eta_\varepsilon(x-y),b(y,\rho_t)\rangle+\lambda\Delta\eta_\varepsilon(x-y))d\rho_t(y)$. Thus, there exists a constant $C_\varepsilon>0$, such that:

$$\sup_{t\in[t_1,t_2],x\in\mathbb{R}^d}\partial_t\varrho^\varepsilon(t,x)\le C_\varepsilon\,.$$

$t\mapsto\nabla\varrho^\varepsilon(t,x)$ is also absolutely continuous by the same reasoning.

### G.3 Proof of Prop. 5

First, we introduce the Talagrand inequality $T_2$.

*Definition* 6. The distribution $\pi$ satisfies the Talagrand inequality $T_2$, if there exists $\alpha>0$ such that for every $\mu\in\mathcal{P}_2(\mathbb{R}^d)$

$$W_2(\mu,\pi)\le\sqrt{\frac{2}{\alpha}D_{\mathrm{KL}}(\mu||\pi)}\,.$$

According to Otto & Villani (2000, Th. 1), LSI implies $T_2$ with the same constant $\alpha$.

In this subsection, we let Assumptions 2, 3 and Assumption 4 hold. Moreover, we assume $\lambda>0$.

Let $\rho\in\mathbb{V}_2$. By Prop. 4 and Assumption 4, we obtain

$$D_{\mathrm{KL}}(\rho_{t_2}||\pi)-D_{\mathrm{KL}}(\rho_{t_1}||\pi)\le-2\alpha\lambda\int_{t_1}^{t_2}D_{\mathrm{KL}}(\rho_t||\pi)dt\,,$$

for every $t_2>t_1>0$. By Grönwall's lemma, we obtain $D_{\mathrm{KL}}(\rho_{t_2}||\pi)\le e^{-2\alpha\lambda(t_2-t_1)}D_{\mathrm{KL}}(\rho_{t_1}||\pi)$. Using the Talagrand inequality $T_2$, we obtain

$$W_2(\rho_{t_2},\pi)\le\sqrt{\frac{2}{\alpha}D_{\mathrm{KL}}(\rho_{t_1}||\pi)}e^{-\alpha\lambda(t_2-t_1)}W_2(\rho_{t_1},\pi)\,,$$

for every $t_2>t_1>0$. Using Eq. (13), the proof is finished.

## H    PROOF OF CONVERGENCE RESULTS

In this section, we let Assumptions 1, 2, and 3 hold. Moreover, we assume $\lambda > 0$.

First, we show the stronger ergodic convegergence result:

**Proposition 7.** *For every sequence $(\varphi_n, \psi_n) \to (\infty, \infty)$, we obtain*

$$\lim_{n \to \infty} \mathbb{P} \left( \frac{\sum_{i \in [\psi_n]} \gamma_i W_2(\mu_i^{\varphi_n}, \pi)}{\sum_{i \in [\psi_n]} \gamma_i} \geq \varepsilon \right) = 0 \,,$$

*for every $\varepsilon > 0$. The latter still holds when we replace $W_2(\cdot, \cdot)$ by $W_2(\cdot, \cdot)^2$.*

*Proof.* By Prop. 1, it is straightforward to check that Bianchi et al. (2024, Cor. 1) holds under Assumptions 1 and 2. The proof consists in identifying the Birkhoff center $\mathrm{BC}_2$, defined hereafter.

We define the translation $\Theta_t : x \in \mathcal{C} \to x(t + \cdot)$. We say that a point $\rho \in \mathbb{V}_2$ is recurrent if there exists a sequence $(t_n)$ such that $\lim_{n \to \infty} (\Theta_{t_n})_{\#} \rho = \rho$. The Birkhoff center $\mathrm{BC}_2$ is the closure of all recurrent points.

Let $\Lambda \subset \mathbb{V}_2$. Let $\mathcal{F} : \mathbb{V}_2 \to \mathbb{R}$ be a l.s.c. function such that $t \mapsto \mathcal{F}((\Theta_t)_{\#} \rho)$ is strictly decreasing when $\rho \notin \Lambda$ and constant when $\rho \in \mathbb{V}_2$. We say that a function $\mathcal{F}$ defined as above is a Lyapunov function for a set $\Lambda$.

**Lemma 9.** *Let $\mathcal{F}$ be a Lyapunov function for a set $\Lambda$. Every recurrent points belongs to $\Lambda$.*

*Proof.* The limit $\ell := \lim_{t \to \infty} \mathcal{F}((\Theta_t)_{\#} \rho)$ is well-defined because $\mathcal{F}((\Theta_t)_{\#} \rho)$ is non increasing. Consider a recurrent point $\rho \in \mathbb{V}_2$, say $\rho = \lim_n (\Theta_{t_n})_{\#} \rho$. Clearly $\mathcal{F}(\rho) \geq \mathcal{F}((\Theta_{t_n})_{\#} \rho) \geq \ell$. Moreover, by lower semi-continuity of $\mathcal{F}$, $\ell = \lim_n \mathcal{F}((\Theta_{t_n})_{\#} \rho) \geq \mathcal{F}(\rho)$. Therefore, $\ell$ is finite, and $\mathcal{F}(\rho) = \ell$. This implies that $t \mapsto \mathcal{F}((\Theta_t)_{\#} \rho)$ is constant. By definition, this in turn implies $\rho \in \Lambda$, which concludes the proof. $\square$

We define the l.s.c. function $\mathcal{F}_{\varepsilon} : \rho \in \mathbb{V}_2 \to D_{\mathrm{KL}}(\rho_{\varepsilon} || \pi)$. By Prop. 4, this is a Lyapunov function for the set

$$\Lambda_{\varepsilon} := \{ \rho \in \mathbb{V}_2 \, : \, \mathcal{I}_{\mathrm{stein}}(\rho_t || \pi) = \mathcal{I}(\rho || \pi) = 0, \, \forall t \geq \varepsilon \, a.e. \} \,.$$

For $\mu \in \mathcal{P}_2(\mathcal{C})$, $\mathcal{I}(\mu || \pi) = 0$ implies $\mu = \pi$, and therefore $D_{\mathrm{KL}}(\mu || \pi) = 0$. Moreover, $t \mapsto D_{\mathrm{KL}}(\rho_t || \pi)$ is constant for $t \geq \varepsilon$. Consequently,

$$\Lambda_{\varepsilon} = \{ \rho \in \mathbb{V}_2 \, : \, \rho_t = \pi, \, \forall t \geq \varepsilon \}.$$

Let $\rho \in \mathbb{V}_2$ a recurrent point, say $\lim_{n \to \infty} (\Theta_{t_n})_{\#} \rho = \rho$. By continuity of the projection $(\pi_0)_{\#}$, we obtain $\lim_{n \to \infty} \rho_{t_n} = \rho_0 = \pi$.

Let $\rho \in \mathrm{BC}_2$. It is a limit of recurrent points $\rho$ satisfying $\rho_0 = \pi$. Hence, still by continuity of the mapping $(\pi_0)_{\#}$, $\rho_0 = \pi$. This finishes the proof of the fist claim of Prop 7.

The second claim holds by the same corollary (Bianchi et al., 2024, Cor. 1). $\square$

Next, we state a stronger convergence result.

**Proposition 8.** *For every sequence $(\varphi_n, \psi_n) \to (\infty, \infty)$, we obtain*

$$\lim_{n \to \infty} \mathbb{P} \left( W_2(\mu_{\psi_n}^{\varphi_n}, \pi) \geq \varepsilon \right) = 0 \,,$$

*for every $\varepsilon \geq 0$.*

*Proof.* By Prop. 5, we obtain

$$\lim_{t \to \infty} \sup_{\rho \in \mathcal{K}} W_2(\rho_t, \pi) = 0 \,, \tag{23}$$

for every compact $\mathcal{K}$ of $\mathcal{P}_2(\mathcal{C})$. Recall that the collection of random variables $\{m_t^n\}$ is tight in $\mathcal{P}_2(\mathcal{C})$ by Lem. 5. Let $(t_n, \varphi_n)$ be a sequence such that $(t_n, \varphi_n) \to_n (\infty, \infty)$ and such that $(m_{t_n}^{\varphi_n})_n$ converges in distribution to $M$. To prove Cor. 8, it will be enough to show that

$$\forall \delta, \varepsilon > 0, \exists T > 0, \quad \limsup_n \mathbb{P} \left( W_2 \left( (\pi_0)_{\#} m_{t_n + T}^{\varphi_n}, \pi \right) \geq \delta \right) \leq \varepsilon.$$

This shows indeed that

$$W_2\left((\pi_0)_{\#}m_t^n, \pi\right) \xrightarrow[(t,n)\to(\infty,\infty)]{\mathbb{P}} 0,$$

and by taking $t = \tau_k$ and by recalling that $(\pi_0)_{\#}m_{\tau_k}^n = \mu_k^n$, we obtain our theorem.

Fix $\delta$ and $\varepsilon$. By the tightness of the family of random variables $\{m_t^n\}$, there exists a compact set $\mathcal{D} \subset \mathcal{P}_2(\mathcal{C})$ such that $\mathbb{P}(m_t^n \in \mathcal{D}) \geq 1 - \varepsilon/2$ for each couple $(t, n)$. This implies that $M(\mathcal{D}) \geq 1 - \varepsilon/2$ by the Portmanteau theorem. Since $\mathbb{V}_2$ is closed by Lem. 6, the set $\mathcal{K} = \mathcal{D} \cap \mathbb{V}_2$ is compact in $\mathcal{P}_2(\mathcal{C})$, and by consequence, it is compact in $\mathbb{V}_2$ for the trace topology. By the same proposition, $M(\mathbb{V}_2) = 1$, therefore, $M(\mathcal{K}) \geq 1 - \varepsilon/2$.

Since $\mathcal{P}_2(\mathcal{C})$ is Polish, we can apply Skorokhod's representation theorem (Billingsley, 1999, Th. 6.7) to the sequence $(m_{t_n}^{\varphi_n})$, yielding the existence of a probability space $(\widetilde{\Omega}, \widetilde{\mathcal{F}}, \widetilde{\mathbb{P}})$, a sequence of $\mathcal{P}_2(\mathcal{C})$–valued random variables $(\rho^n)$ on $\widetilde{\Omega}$ and a $\mathcal{P}_2(\mathcal{C})$–valued random variable $\rho^\infty$ on $\widetilde{\Omega}$ such that $(\rho^n)_{\#}\widetilde{\mathbb{P}} = (m_{t_n}^{\varphi_n})_{\#}\mathbb{P}$, $(\rho^\infty)_{\#}\widetilde{\mathbb{P}} = M$, and $\rho^n \to \rho^\infty$ pointwise on $\widetilde{\Omega}$. Noting that $(\pi_0)_{\#}m_{t_n+T}^{\varphi_n}$ and $\rho_T^n$ have the same probability distribution as $\mathcal{P}_2(\mathbb{R}^d)$–valued random variables, we show that

$$\exists T > 0, \quad \limsup_n \widetilde{\mathbb{P}}\left(W_2\left(\rho_T^n, \pi\right) \geq \delta\right) \leq \varepsilon, \tag{24}$$

to establish our theorem. Applying Eq. (23) to the compact $\mathcal{K}$, we set $T > 0$ in such a way that

$$\sup_{\rho \in \mathcal{K}} W_2(\rho_T, \pi) \leq \delta/2.$$

By the triangular inequality, we have

$$W_2\left(\rho_T^n, \pi\right) \leq W_2\left(\rho_T^n, \rho_T^\infty\right) + W_2\left(\rho_T^\infty, \pi\right).$$

The first term at the right hand side converges to zero for each $\tilde{\omega} \in \widetilde{\Omega}$ by the continuity of the function $\rho \mapsto \rho_T$, thus, this convergence takes place in probability. We also know that for $\widetilde{\mathbb{P}}$–almost all $\tilde{\omega} \in \widetilde{\Omega}$, it holds that $\rho^\infty \in \mathbb{V}_2$. Thus, regarding the second term, we can write

$$\widetilde{\mathbb{P}}\left(W_2\left(\rho_T^\infty, \pi\right) \geq \delta\right) \leq \widetilde{\mathbb{P}}\left(\rho^\infty \notin \mathcal{K}\right) + \widetilde{\mathbb{P}}\left((W_2\left(\rho_T^\infty, \pi\right) \geq \delta) \cap (\rho^\infty \in \mathcal{K})\right).$$

When $\rho^\infty \in \mathcal{K}$, it holds that $W_2\left(\rho_T^\infty, \pi\right) \leq \delta/2$, thus, the second term at the right hand side of the last inequality is zero. The first term satisfies $\widetilde{\mathbb{P}}\left(\rho^\infty \notin \mathcal{K}\right) = 1 - M(\mathcal{K}) \leq \varepsilon/2$, and the statement (24) follows. Cor. 8 is proven. $\qquad\square$

### H.1 PROOF OF TH. 2

Instead of seeing $\overline{\mathscr{L}}^n$ as set of random variable on $\mathcal{P}_2(\mathbb{R}^d)$, we see it as a set of measures in $\mathcal{P}(\mathcal{P}_2(\mathbb{R}^d))$. We denote such a set as $\overline{\mathcal{L}}^n$.

Let $\varepsilon > 0$. By contradiction, there exists $\delta > 0$, a subsequence $\varphi_n \to \infty$ and a sequence of measures $\nu^n \in \overline{\mathcal{L}}^{\varphi_n}$ satisfying

$$\int \mathbf{1}_{W_2(\mu,\pi)>\varepsilon} d\nu^n(\mu) \geq \delta.$$

As shown in the proof of Th. 1, the sequence of random variable $(\bar{\mu}_k^n : k, n \in \mathbb{N}^*)$ is tight. Hence, there exists a measure $\nu^\infty \in \mathcal{P}_2(\mathbb{R}^d)$ such that $(\nu^n)$ converges to $\nu^\infty$ along a subsequence. To keep the notations simple, we say that $\nu^n \to \nu^\infty$. Since, $\mu \in \mathcal{P}_2(\mathbb{R}^d) \mapsto \mathbf{1}_{W_2(\mu,\pi)}$ is continuous bounded, we obtain

$$\int \mathbf{1}_{W_2(\mu,\pi)>\varepsilon} d\nu^\infty(\mu) \geq \delta.$$

Let $(\psi_k^n)_k$ be a sequence diverging to $\infty$ such that $\bar{\mu}_{\psi_k^n}^n \to_k \nu^n$, for every $n \in \mathbb{N}^*$.

Let $\varepsilon' > 0$, there exists $n_0$ such that,

$$\left|\int \mathbf{1}_{W_2(\mu,\pi)>\varepsilon} d\nu^\infty(\mu) - \int \mathbf{1}_{W_2(\mu,\pi)>\varepsilon} d\nu^{n_0}(\mu)\right| \leq \frac{\varepsilon'}{2}.$$

Moreover, there exists $k_0$ such that

$$\left| \mathbb{P}(W_2(\bar{\mu}_{\psi_{k_0}^{n_0}}^{n_0}, \pi) > \varepsilon) - \int \mathbf{1}_{W_2(\mu, \pi) > \varepsilon} d\nu^{n_0}(\mu) \right| \leq \frac{\varepsilon'}{2} \,.$$

Consequently, there exists a subsequence $(\tilde{\varphi}_n, \tilde{\psi}_n) \to (\infty, \infty)$ such that

$$\lim_{n \to \infty} \mathbb{P}(W_2(\bar{\mu}_{\tilde{\psi}_n}^{\tilde{\varphi}_n}, \pi) \geq \varepsilon) = \int \mathbf{1}_{W_2(\mu, \pi) > \varepsilon} d\nu^{\infty}(\mu) \geq \delta \,.$$

By Jensen's inequality, we obtain

$$W_2(\bar{\mu}_{\tilde{\psi}_n}^{\tilde{\varphi}_n}, \pi)^2 \leq \frac{\sum_{k \in [\tilde{\psi}_n]} \gamma_k W_2(\mu_k^{\tilde{\varphi}_n}, \pi)^2}{\sum_{k \in [\tilde{\psi}_n]} \gamma_k} \,.$$

Consequently,

$$\lim_{n \to \infty} \mathbb{P}\left( \frac{\sum_{k \in [\tilde{\psi}_n]} \gamma_k W_2(\mu_k^{\tilde{\varphi}_n}, \pi)^2}{\sum_{k \in [\tilde{\psi}_n]} \gamma_k} \geq \varepsilon^2 \right) \geq \delta \,.$$

The latter contradicts the second claim of Prop. 7. Thus, the proof is finished.

### H.2 Proof of Th. 3

This is the same proof as Th. 2. But this time, we use Prop. 8 instead of Prop. 7.

### H.3 Proof of Cor. 1

By contradiction, assume that there exists $\delta > 0$ and a subsequence $\varphi_n$, such that for every $n$, $\limsup_{k \to \infty} \mathbb{P}(W_2(\bar{\mu}_k^{\varphi_n}, \pi) \geq \varepsilon) > \delta$. Assume $\varphi_n = n$ to simplify the notations. For any $n$, this implies that one can extract a subsequence, say $(\psi_k^n : k \in \mathbb{N})$, such that for every $k$, $\mathbb{P}(W_2(\bar{\mu}_{\psi_k^n}^n, \pi) \geq \varepsilon) > \delta/2$. By Th. 1, the sequence $(\bar{\mu}_{\psi_k^n}^n : k \in \mathbb{N})$ is tight, so that there exists $\nu^n \in \overline{\mathscr{L}}^n$, such that $\bar{\mu}_{\psi_k^n}^n$ converges in distribution to $\nu^n$ as $k \to \infty$, along some subsequence which we still denote by $\psi_k^n$ to keep the notations simple. By the Portmanteau theorem,

$$\limsup_{k \to \infty} \mathbb{P}(W_2(\bar{\mu}_{\psi_k^n}^n, \pi) \geq \varepsilon) \leq \mathbb{P}(W_2(\nu^n, \pi) \geq \varepsilon) \,. \tag{25}$$

By Th. 2, $\nu^n$ converges in probability to $\pi$ in $\mathcal{P}_2(\mathbb{R}^d)$ as $n \to \infty$. Therefore, $\mathbb{P}(W_2(\nu^n, \pi) \geq \varepsilon) < \delta/3$ for all $n$ large enough. Using Eq. (25), it follows that $\mathbb{P}(W_2(\bar{\mu}_{\psi_k^n}^n, \pi) \geq \varepsilon) < \delta/2$ along some subsequence, hence a contradiction. This proves the first point. The second point follows the same arguments.

