# OpenReview forum: "Long-time asymptotics of noisy SVGD outside the population limit"
_ICLR.cc/2025/Conference — ICLR 2025 Poster_

### Official Review · Reviewer_vKXm · 2024-10-27

**Soundness:** 2
**Presentation:** 3
**Contribution:** 2
**Rating:** 6
**Confidence:** 3

**Summary:**

The paper poses a question on Stein variational gradient descent:
>What does SVGD converge to (i.e., as k → ∞) in the finite particles regime (i.e., when n < ∞ is
fixed)?

The paper then introduces a noisy version of SVGD (NSVGD), where each particle simultaneously undergoes Langevin regularization with regularization strength $\lambda \geq 0$.  In the case of $\lambda =0$, this recovers SVGD

The paper then proves convergence of the NSVGD under a vanishing stepsize setup ($\sum \gamma_k = \infty$ and $\gamma_k \downarrow 0$).  Under sufficient regularity on the kernel K (Assumption 3), this holds for the averaged iterate (Theorem 3), and under an additional log-soblev inequality assumption on the target measure $\pi$, this holds for the final iterate.

The paper is principally concerned with proof, and an overview of the proof is given in Section 5 of the main-text.  This builds on existing work around McKean-Vlasov SDEs and optimal transport theory.

Finally, in Section 6, the article provides numerical simulations illustrating that NSVGD avoids 'particle collapse', where the particles in SVGD can 'freeze' in high-dimensions.  This is measured by computing the average variance of the coordinates sampled from a high-dimensional distribution.  When dimension $d$ is large with respect to the number of particles,  (indeed from figures, this would appear to be in a proportionate $d \propto n$ regime), a non-degenerate gap appears between this variance and the ground truth, whereas numerical simulations suggest that for NSVGD this does not occur.

**Strengths:**

The paper's primary contributions are its main Theorems 2,3.  These are very clearly presented with a good level of mathematical detail.

The paper introduces an algorithm, with improved convergence characteristics with respect to SVGD.  It appears to be a natural extension of SVGD, with essentially identical complexity to SVGD and improved characteristics.

**Weaknesses:**

There are a few overall weaknesses.

1) The paper poses a question on SVGD, but then in fact does not answer this question (What does SVGD, $\lambda= 0$, converge to?), nor does the bulk of the paper seem particularly interested in answering it.  On the contrary, it would seem that SVGD has some obvious (empirically demonstrated) problems with convergence (especially the variance collapse), and you have found a better algorithm.  I think this is mostly a weakness of presentation, and it could be resolved by describing better what will appear in the paper (i.e. mostly not posing this question in italics).  Or alternatively, it should be addressed, somehow?

2) The variance collapse effect in Section 6 leaves open a deeper theoretical evaluation of SVGD, in particular proving a theorem which demonstrates this variance collapse effect for SVGD.  Furthermore, it is not clear if either the NSVGD theory proven would be enough to demonstrate this non-collapse (though in the Gaussian kernel case perhaps it is close?)

3) Since this is a new algorithm, it would be nice to see another panel of performance evaluations, comparing NSVGD to SVGD and Langevin sampling for a standard class of metrics, to get some sense to what extent there is a tradeoff in using something like NSVGD.  For example, to the extent that lambda > 0 acts like a regularizer, one may expect that too much regularization causes performance losses (on the other hand, given that it comes with the Langevin generator, perhaps this never occurs?)

**Questions:**

Questions related to the weaknesses:

Addressing Weakness 2:  Do the theorems allow any uniformity in the dimension d of the underlying problem, say in the case of a log-sobolev constants?

Addressing Weakness 3:  Can you provide further experiments, evaluating performance of NSVGD vs SVGD and Langevin sampling?  Is there any setup, in particular where lambda=0 is favorable?

Other questions:
1) On l72 you say:
> Indeed, SVGD does not converge to the target distribution when $n<\infty$. This is because the iterates of SVGD are discrete measures with a finite support of $n$ points, whereas the target $\pi$ has a continuous density with respect to the Lebesgue measure.

But the _law_ of those $n$ points may very well have a density, and moreover the law of a randomly selected point could be close to $\pi$.  Why is the discreteness of this measure a problem? (You also mention this on l84). Furthermore, even if this were a problem, shouldn't one just change the metric?
2)  On l.358:
>Moreover, in the population limit where $n$ is large, any of the continuous-time particles coincides, in law, with the solution to a McKean-Vlasov equation, as defined below. This phenomenon is known as the propagation of chaos. We refer to Chaintron \& Diez (2022) for a detailed exposition.

Why should _all_ particles need converge, instead of the evolution of a _tagged_ particle converges? (For example the right-most particle in a 1-dimensional setup need not converge)

3) l.386.  There is some double limit notation not properly introduced.

4) l.517.  ``usefull''

---

> ### Author Response · Authors · 2024-11-21
>
> Thank you for your positive review and thoughtful questions. Please find our responses below.
>
> *The paper poses a question on SVGD, but then in fact does not answer this question (What does SVGD, $\lambda=0$, converge to?), nor does the bulk of the paper seem particularly interested in answering it.*
>
> When $ \lambda = 0 $, the algorithm converges to a set of measures that includes the target $ \pi $ as well as some probability measures without density (the addition of the noise allows to avoid measure witout density). However, you are correct that we should not have emphasized this point. As you suggested, we have removed the italics in the updated version.
>
>
> Indeed, a large noise ($\lambda \gg 1$) will not bias the algorithm. As demonstrated in the newly added Section C, by setting $\lambda$ large, NSVGD behaves like the Langevin algorithm. In fact, it effectively becomes a Langevin algorithm with a step size of $\lambda\gamma\_k$. Therefore, to ensure convergence, it is necessary to have $\lambda\gamma\_k \ll 1$.
>
> *Do the theorems allow any uniformity in the dimension d of the underlying problem, say in the case of a log-sobolev constants?*
>
> In our proof technique, we cannot establish bounds that provide a definitive answer. However, we can state that the number of particles ($n$) should depend exponentially on the dimension. Indeed, the larger the space, the more particles are needed to adequately cover it. Moreover, the time has a linear dependence on the dimension. Specifically, by examining the proof of Lemma 7, we observe that the constant $C$ (which serves as an upper bound for the Kullback-Leibler divergence) depends linearly on $d$. Additionally, the log-Sobolev constant corresponds to the convexity constant of $F$, where we recall that the target distribution is proportional to $\exp(-F)$.
>
> To summarize, we believe there exists a strong dependency on the dimension $d$.
>
>
> This version corrects grammatical issues and improves the clarity and flow of the text.
> *Can you provide further experiments, evaluating performance of NSVGD vs SVGD and Langevin sampling? Is there any setup, in particular where lambda=0 is favorable?*
>
> We hope to satisfy your requirements in Section C of the updated version of our manuscript. To summarize, we used the toy example of a Neal funnel density. In this setup, SVGD outperforms the Langevin algorithm. We show that by setting $\lambda$ small enough, NSVGD retains the performance of SVGD. Therefore, NSVGD is advantageous as it performs similarly to SVGD and better than Langevin, while also providing convergence guarantees. But, in this particular setup, $\lambda=0$ yields better performance.
>
> *The law of those $n$ points may very well have a density, and moreover the law of a randomly selected point could be close to $\pi$. Why is the discreteness of this measure a problem? (You also mention this on l84). Furthermore, even if this were a problem, shouldn't one just change the metric?*
>
> By stating that SVGD converges to $ \pi $, we mentioned in line 51 that this means the empirical measure $\mu\_k^n $ converges to  $\pi $. In this case, for SVGD, one can only hope for
> $ n \to \infty $.
>
>
> As you said, one could alternatively define the convergence of SVGD as the law of a single particle converging to $ \pi $.
> But, SVGD works because, in the population limit $ n = \infty $, the Kullback–Leibler divergence $ D\_{KL}(\mu\_k^\infty \| \pi) $ decreases along $ k $. This property cannot hold for finite $ n $, since $ D\_{KL}(\mu \| \pi) $ is defined only for measures $ \mu $ admitting a density. To the best of our knowledge, $ D\_{KL}(\text{Law}(X\_k^{1,n}) \| \pi) $ does not decrease for $n<\infty$.
>
> *Why should all particles need converge, instead of the evolution of a tagged particle converges?*
>
> By Assumption 1-(ii), the particles at time $k =0$ share the same distribution. Furthermore, by the definition of the algorithm, we can verify that all particles $X\_k^{1,n}, \dots, X\_k^{n,n}$ also share the same distribution at any time $k$. Consequently, we can only expect all the particles to converge or none at all.
>
> *l.386. There is some double limit notation not properly introduced, l.517. ``usefull''*
>
> Thank you for your careful reading. This has now been corrected in the updated version (the definition of the double limit is in l.139.).
>
> **We agree with you that it would be interesting to establish a convergence result when the dimension $ d $ grows with $ k $ and $ n $. However, at present, we are unable to establish such a result. We hope we have clarified the reason why it is necessary to study the set of all particles rather than focusing on a single selected particle. Additionally, we hope the new Section C in the updated manuscript provides clarity on the behavior of NSVGD compared to SVGD. If you are satisfied with our response, we would be deeply grateful if you would consider raising your score.**

---

> > ### Comment · Reviewer_vKXm · 2024-11-23
> >
> > Thank you for the response.  I have no additional questions.
> >
> > I'll reiterate that I'm overall supportive of publication, and I look forward to discussing the paper with the reviewers and AC.

---

### Official Review · Reviewer_WjWs · 2024-10-28

**Soundness:** 4
**Presentation:** 4
**Contribution:** 4
**Rating:** 8
**Confidence:** 4

**Summary:**

This paper studies the long-time behavior of noisy SVGD, by first taking the limit to infinity in the number of steps, and then taking the limit to infinity in the number of particles. Noisy SVGD is a variant of the SVGD algorithm where a Langevin update is added to the SVGD update. Using propagation of chaos arguments, the authors are able to show that the distributional limit set of NSVGD at a finite number of particles $n$ converges in probability to the target distribution $\pi(x) \propto \exp(-F(x))$ as $n$ goes to infinity. Since the limit in which the number of particles goes to infinity first and the number of steps goes second can be understood by the arguments from previous works, that implies that the order of the limits can be exchanged.

Experimentally, the authors show that for two different kernels, different number of particles, and increasing dimension $d$, the dimension-averaged marginal variance is much lower than it should for SVGD, while NSVGD performs correctly.

**Strengths:**

- The paper resolves a well-known issue of SVGD (variance collapse) by coming up with a modification that provably and empirically does not show variance collapse. The results of the paper are solid both in theory and practice.
- The theory of the paper relies heavily on results on McKean-Vlasov equations, which are not part of the standard toolkit for machine learning researchers. This paper can serve as a guide to finite-particle convergence results for other particle algorithms.

**Weaknesses:**

- The paper would be really complete if the authors gave some guidance on how to appropriately set the Langevin regularization parameter, at least in the settings that they have studied. That would be useful for practitioners looking to replace SVGD or the Langevin algorithm by NSVGD.

**Questions:**

- Figure 1 is illustrative of the fact that NSVGD does not suffer from variance collapse, while regular SVGD does. In that sense, it is a correct validation of the theory presented in the paper. However, it would also be interesting to compare empirically NSVGD to Langevin dynamics. That is, is the SVGD term in NSVGD actually useful, or is the reason for the success of NSVGD simply the Langevin term? Should practitioners use NSVGD or just Langevin? In my view, resolving this issue is critical for NSVGD to be considered a useful algorithm. I’ll be happy to raise my score if this is addressed.
- While introducing the Langevin terms is useful to fix variance collapse, do the authors know if NSVGD is uniformly better than SVGD? For example, oftentimes SDEs may require a larger number of steps than ODEs to obtain numerical errors below a certain threshold. Can the authors comment on that?
- Typos: In line 246, SGVD -> NSVGD. In line 517, usefull -> useful.

---

> ### Author Response · Authors · 2024-11-21
>
> We thank you for your positive review and your kind comments. Please find our responses below.
>
> *Figure 1 is illustrative of the fact that NSVGD does not suffer from variance collapse, while regular SVGD does. In that sense, it is a correct validation of the theory presented in the paper. However, it would also be interesting to compare empirically NSVGD to Langevin dynamics. That is, is the SVGD term in NSVGD actually useful, or is the reason for the success of NSVGD simply the Langevin term? Should practitioners use NSVGD or just Langevin? In my view, resolving this issue is critical for NSVGD to be considered a useful algorithm. I’ll be happy to raise my score if this is addressed.*
>
> We address your concern in Section C of the updated version of our manuscript. To summarize, we used the toy example of a Neal funnel density. In this setup, SVGD outperforms the Langevin algorithm. We show that by setting $\lambda$ small enough, NSVGD retains the performance of SVGD. Therefore, NSVGD is advantageous as it performs similarly to SVGD and better than Langevin, while also providing convergence guarantees.
>
> *While introducing the Langevin terms is useful to fix variance collapse, do the authors know if NSVGD is uniformly better than SVGD? For example, oftentimes SDEs may require a larger number of steps than ODEs to obtain numerical errors below a certain threshold. Can the authors comment on that?*
>
> It is true that the Euler scheme for an SDE often requires a smaller step size than the Euler scheme for an ODE. This could be another reason why SVGD may perform better than the Langevin algorithm in some cases.
>
> We did not address the convergence speed of the algorithm. However, it is worth noting that in stochastic approximation, it is common to introduce noise into an algorithm. In cases where the algorithm performs well, adding noise may decrease its performance. However, in more challenging cases, noise can help the algorithm explore the space more effectively and avoid getting trapped in local minima.
>
> In NSVGD, the Langevin regularization plays a similar role. As you pointed out, since SDEs are harder to approximate than ODEs, this regularization can slow down the algorithm in cases where it already performs well. However, in more challenging scenarios, such as high-dimensional problems ($d \gg 1$), SVGD alone may struggle, as the particles tend to cluster and the algorithm can get stuck in a trap. In such cases, NSVGD's regularization can provide an advantage by avoiding these issues.
>
> To answer your question, NSVGD is not uniformly better than SVGD but has better performance in certain challenging cases.
>
> *Typos: In line 246, SGVD -> NSVGD. In line 517, usefull -> useful.*
>  Thank you for your careful reading. We have addressed this in the updated version of the manuscript.
>
>
> **We hope that the new Section C in the updated version highlights the value of NSVGD compared to Langevin. If you are satisfied with our experiments and explanations regarding the role of Langevin noise, we would be grateful if you could consider raising your score.**

---

> > ### Comment · Reviewer_WjWs · 2024-11-26
> > **Reply to authors' response**
> >
> > I am satisfied by the authors' response, and I raised my score.

---

### Official Review · Reviewer_SEGH · 2024-10-30

**Soundness:** 2
**Presentation:** 2
**Contribution:** 2
**Rating:** 5
**Confidence:** 4

**Summary:**

The paper shows that noisy SVGD can properly sample a target distribution.

**Strengths:**

The paper shows that noisy SVGD can properly sample a target distribution.

**Weaknesses:**

The results are not all that surprising, and I'm surprised earlier papers haven't proved this before. The background section may not have been thorough enough.

**Questions:**

Why didn't earlier papers prove such results before?

If indeed the results are novel, what's the main barrier for theoretical analysis that was overcome?

---

> ### Author Response · Authors · 2024-11-21
>
> We thank you for your review.  Please find our responses below.
>
> *Why didn't earlier papers prove such results before?*
>
> Convergence of the SVGD algorithm when the time grows can only be rigorously established in the population limit $ n = \infty $. Therefore, to analyze the convergence of the SVGD algorithm in the finite particle regime, we must show that the finite particle system converges to the population limit as $ n \to \infty $. This property is known as propagation of chaos.
>
> A well-known challenge with propagation of chaos is that it typically holds only on a finite time window, due to the use of Grönwall’s inequality. As a consequence, with this type of result, the convergence of SVGD can only be guaranteed if the number of particles grows exponentially with the number of iterations.
>
> *If indeed the results are novel, what's the main barrier for theoretical analysis that was overcome?*
>
> Instead of using the classical approach based on coupling, we employed a method relying on a tightness argument, specifically designed to study the long-time convergence of particle systems. However, a significant challenge with our approach is that it does not guarantee the avoidance of spurious stationary measures. Since such spurious stationary measures are observed in practice, we introduce a noise term to fix SVGD and ensure convergence to the desired distribution.
>
> **We hope to have clarified the importance of this work. SVGD is a well-known algorithm in the variational inference literature, liked for its elegant mathematical properties and strong performance across various applications. We believe we have made significant progress in addressing an important question: "Does SVGD converge?" We would be happy to answer any further questions, and we would greatly appreciate it if you would consider raising your score.**

---

### Official Review · Reviewer_tkfd · 2024-11-04

**Soundness:** 3
**Presentation:** 3
**Contribution:** 2
**Rating:** 5
**Confidence:** 2

**Summary:**

This paper studies the convergence of a noisy version of SVGD.
The noisy version of SVGD in this paper is defined as a mixture of SVGD and Langevin dynamics.
Specifically, a small Langevin dynamics update is added to each SVGD update.
The authors prove the convergence of this noisy SVGD.
In particular, they demonstrate that the empirical distribution of the particles converges to the target distribution in probability, under the condition that the number of iterations goes to infinity, followed by the number of particles going to infinity.
Empirical evaluations demonstrate that the noisy SVGD avoids variance collapse, unlike vanilla SVGD.

**Strengths:**

The theoretical understanding of SVGD has been lacking over the years.
In particular, most existing convergence analyses are based on population limits.
Instead, this paper attempts to study the asymptotic convergence behavior of finite particles, specifically by letting the number of iterations go to infinity and then letting the number of particles go to infinity.
A step in this direction helps deepen the understanding of SVGD.

**Weaknesses:**

1. The main weakness is that the noisy version of SVGD deviates a lot from the SVGD used in practice.
Presumably, the main reason for analyzing this particular version of noisy SVGD is to resolve certain technical challenges in the proof.
1. The coefficient of the Langevin dynamics controls how similar the noisy SVGD is to vanilla SVGD and Langevin dynamics.
Large coefficients reduce to Langevin dynamics, whereas small coefficients reduce to vanilla SVGD.
It is not clear whether there is a sweet spot where a moderate coefficient is better than both vanilla SVGD and Langevin dynamics.

**Questions:**

1. Is it possible to prove noisy SVGD with other types of noise?
What would be the technical challenges there?

---

> ### Author Response · Authors · 2024-11-21
>
> Thank you for the time you spent on the paper. Please find our responses below.
>
> *The main weakness is that the noisy version of SVGD deviates a lot from the SVGD used in practice.*
>
> We have added a new experiment in Section C of the updated manuscript. In this section, we demonstrate that by setting $\lambda$ sufficiently small, NSVGD exhibits the same dynamics as SVGD.
>
> *Presumably, the main reason for analyzing this particular version of noisy SVGD is to resolve certain technical challenges in the proof.*
>
> The noise not only resolves certain technical issues in the proof but also addresses a major problem of SVGD: mode collapse (see Section 6).
>
>
> *Is it possible to prove noisy SVGD with other types of noise? What would be the technical challenges there?*
>
>
>
> We chose to add noise of the form $-\lambda\gamma\_{k+1} \nabla F(X^{i,n}\_k) + \sqrt{2\lambda\gamma\_{k+1}} \xi^{i,n}\_k$, where $(\xi\_k^{i,n})\_{i, k}$ are i.i.d. centered Gaussian random variables with variance 1. However, $\xi\_k^{i,n}$ could, in fact, be any i.i.d. random variable with variance 1, zero mean, and finite fourth-order moments. The proof would remain the same since the limiting McKean-Vlasov equation would be the same.
>
> Alternatively, we could consider noise of the form $\sqrt{2\gamma\_{k+1}\lambda} \xi^{i,n}\_k$, where $\xi^{i,n}\_k$ is a centered random variable with variance 1 with finite fourth-order moments. However, such noise would bias the algorithm, causing convergence to a set $S\_\lambda$ that depends on $\lambda$. As $\lambda \to 0$, $S\_\lambda$ would converge (in the sense of the Hausdorff distance) to the singleton $\{\pi\}$.
>
> **We hope that the new Section C in the updated version demonstrates that we have proven the convergence of an algorithm whose behavior resembles SVGD. In proving the convergence of an algorithm whose dynamics resemble those of SVGD, we believe we have made significant progress in understanding the convergence properties of SVGD. We also believe that SVGD requires some noise to ensure convergence. While various types of noise could have been considered, in our manuscript, we chose the simplest form of noise that does not bias the algorithm. We hope this addresses your concerns, and if so, we would greatly appreciate it if you could consider raising your score.**

---

### Author Response · Authors · 2024-11-21

We have uploaded a new version of our manuscript with the following updates:

- A new section in the appendix (Section C) that includes additional simulations requested by the reviewer. In this section, we present a toy example where SVGD  outperforms the Langevin algorithm, and NSVGD achieves the same performance as SVGD . This section is referenced in line 523.

- The addition of a missing definition in line 139, as well as corrections to typos in lines 517 and 246.

Moreover, we have added the code which has produced Section C in the supplementary.

---

### Meta-Review · Area_Chair_1Wg6 · 2024-12-21

**Metareview:**

This paper studies the long-time asymptotics of a noisy variant of SVGD. While the algorithm is not very common,
authors characterize  its limit set for large number of iterations and show that it approaches the target with more particles. Interestingly, they also show this algorithms can avoid variance collapse which is a problem for SVGD.

This paper was reviewed by three reviewers the following Scores/Confidence: 6/3, 5/2, 8/4. I think the paper is studying an interesting topic and the results are relevant to ICLR community. The following concerns were brought up by the reviewers:

- Authors should add discussion on uniformity in the dimension d, e.g. in the case of a log-sobolev constants

- I think the variance collapse statement in the abstract can be clarified, either with reference or numerics. Is there a rigorous theorem in the same setting showing this for SVGD? Can the authors provide a separation result for SVGD and noisy version considered in this paper? I am not asking for new results, but the statements should be more precise.

Authors should carefully go over reviewers' suggestions and address any remaining concerns in their final revision. Based on the reviewers' suggestion, as well as my own assessment of the paper, I recommend including this paper to the ICLR 2025 program.

**Additional Comments On Reviewer Discussion:**

Reviewer questions are thoroughly answered by the authors.

---

### Decision · Program_Chairs · 2025-01-22

Accept (Poster)